# Gene therapy for epilepsy targeting neuropeptide Y and its Y2 receptor to dentate gyrus granule cells

Stefano Cattaneo [1,2,5], Barbara Bettegazzi [1,2,5], Lucia Crippa [1,2], Laila Asth[3], Maria Regoni[1,2], Marie Soukupova [3], Silvia Zucchini [3], Alessio Cantore [1,4], Franca Codazzi[1,2], Flavia Valtorta [1,2] & Michele Simonato [2,3 ✉]

## Abstract

**Gene therapy is emerging as an alternative option for individuals with drug-resistant focal epilepsy. Here, we explore the potential of a novel gene therapy based on Neuropeptide Y (NPY), a well-known endogenous anticonvulsant. We develop a lentiviral vector co-expressing NPY with its inhibitory receptor Y2 in which, for the first time, both transgenes are placed under the control of the minimal CamKIIa(0.4) promoter, biasing expression toward excitatory neurons and allowing autoregulation of neuronal excitability by Y2 receptor-mediated inhibition. Vector-induced NPY and Y2 expression and safety are first assessed in cultures of hippocampal neurons. In vivo experiments demonstrate efficient and nearly selective overexpression of both genes in granule cell mossy fiber terminals following vector administration in the dentate gyrus. Telemetry video-EEG monitoring reveals a reduction in the frequency and duration of seizures in the synapsin triple KO model. This study shows that targeting a small subset of neurons (hippocampal granule cells) with a combined overexpression of NPY and Y2 receptor is sufficient to reduce the occurrence of spontaneous seizures.**

**Keywords** Glutamate; Hippocampus; Mouse; Synapsin
**Subject Categories** Genetics, Gene Therapy & Genetic Disease; Neuroscience

## Introduction

Epilepsy is the most common serious disorder of the brain, affecting 50–60 million people worldwide (WHO. Epilepsy. Key facts. https://www.who.int/news-room/fact-sheets/detail/epilepsy). Despite significant advances in knowledge and development of therapies, more than 30% of the patients are drug-resistant, i.e., have inadequate seizure control with pharmacological therapy (Kwan and Brodie, 2000; Kwan et al, 2009). This condition is associated with high rates of depression, suicide, and social exclusion. Unfortunately, these figures have not been modified by the introduction of new drugs over the past two decades (Löscher and Schmidt, 2011), and non-pharmacological therapies, like surgery, ketogenic diet, vagal nerve or deep brain stimulation, are applicable to limited cases or have limited efficacy (West et al, 2019; Panebianco et al, 2015; Sprengers et al, 2017; Martin-McGill et al, 2018). Therefore, the identification of alternative, effective treatment options is highly urgent.

Within this scenario, gene therapy is emerging as a doable strategy, based on success in clinical applications for other neurological disorders and on developments in the epilepsy field that are leading to the first clinical trials (Morris and Schorge, 2022; Bettegazzi et al, 2024). In genetic forms of epilepsy, where the disease is caused by a known gene mutation, the most obvious strategy is delivering functional copies of the defective gene (Jensen et al, 2021). However, many cases of epilepsy lack obvious causative mutations. In these situations, gene therapy can be based on delivering genes that address the mechanism(s) of seizure generation (Morris and Schorge, 2022). Such a "mechanistic" approach has been pursued by transferring genes that can modify cell function and control hyperexcitability, like ion channels, neurotransmitters, neurotrophic factors, or receptors (Ingusci et al, 2019).

Irrespective of the underlying cause, seizures represent excessive synchronized discharges of neurons and require the recruitment of excitatory neurons to propagate. In fact, many of the mechanistic gene therapy strategies mentioned above hold broad-spectrum effects and proved effective in multiple epilepsy models, including some of genetic etiology (Street et al, 2023). With rare exceptions (Qiu et al, 2022), however, these approaches did not include a built-in auto-regulatory system, and, therefore, may lead to excessive inhibition.

Representative examples in this respect are the gene therapies based on Neuropeptide Y (NPY), a neuropeptide physiologically released by GABAergic interneurons. The action of NPY in the central nervous system is mediated by receptors (mainly Y1, Y2, and Y5) that are expressed by distinct neuronal sub-populations in several areas of the brain and are localized either pre- or post-synaptically. This gives rise to a complex system with a predominant but not exclusive inhibitory profile, the anti-seizure

[1]Vita-Salute San Raffaele University, 20132 Milan, Italy. [2]Division of Neuroscience, IRCCS San Raffaele Scientific Institute, 20132 Milan, Italy. [3]Department of Neuroscience and Rehabilitation, University of Ferrara, 44121 Ferrara, Italy. [4]San Raffaele Telethon Institute for Gene Therapy, IRCCS San Raffaele Scientific Institute, 20123 Milan, Italy. [5]These authors contributed equally: Stefano Cattaneo, Barbara Bettegazzi. ✉E-mail: simonato.michele@hsr.it

effects being mediated by Y2 and Y5, but not by Y1, receptors (Cattaneo et al, 2021). Many pre-clinical studies in multiple genetic and induced models of epilepsy have tested NPY-based gene therapy products (Cattaneo et al, 2021). Most of the initial approaches used NPY-expressing AAV vectors directly infused in epileptogenic areas. More recently, Y2 or Y5 receptors were also overexpressed, either alone or in combination with NPY, with the aim of re-shaping the response to favor inhibition (Woldbye et al, 2005) and, ultimately, the combined overexpression of NPY and the Y2 receptor (Y2R) proved to be the most effective approach (Melin et al, 2019, 2023). The downsides of all these studies, however, were the lack of specificity for excitatory neurons and the lack of autoregulation. In fact, attempts to obtain NPY-Y2R overexpression used ubiquitous (Szczygieł et al, 2020a) or neuron-specific (Nikitidou Ledri et al, 2016), but not excitatory neuron-specific promoters. Lack of specificity may be problematic because inhibition of inhibitory neurons may increase excitability and promote seizures. Lack of feedback autoregulation may lead to excessive inhibition and compromise function.

In an attempt to circumvent these problems, we designed a lentiviral vector (LV) platform in which NPY and its Y2R are both expressed by a single vector, under the control of a promoter that is mainly active in excitatory cells (the minimal CamKIIa, mCamKIIa, promoter). The choice of using lenti- rather than AAV vectors is primarily based on the aim of restricting the expression, for what possible, to dentate gyrus granule cells (DG GCs), a specific population of excitatory, glutamatergic neurons that is very often removed during epilepsy surgery, because they are considered a "gate" to hippocampal over-excitation, and preventing GC hyperexcitability can be sufficient to prevent the occurrence of spontaneous seizures (Krook-Magnuson et al, 2015). In this respect, the greater spread of AAV vectors was viewed as a disadvantage. In addition, the integration competence of LV vectors guarantees a constant expression of the transgenes, that, in a therapeutic perspective, would also be advantageous as it eliminates the need of repeated administrations.

This approach differs from those used thus far, that employed promoters that are active in all neurons (Melin et al, 2019; Szczygieł et al, 2020a). On one hand, as mentioned above, the mCamKIIa promoter is expected to bias transgene expression toward excitatory neurons, highly limiting their expression in inhibitory neurons; on the other hand, since both transgenes are expressed together in each transduced neuron, it is expected the formation of an auto-inhibitory feedback. Hyperactivity of transduced excitatory neurons should lead to the release of NPY and inhibition of synaptically connected cells; at the same time, expression of Y2R by the very same NPY-releasing neurons should lead to feedback auto-inhibition whenever neuronal activity becomes excessive. This auto-regulatory mechanism differs from previous auto-regulated gene therapies for epilepsy. Lieb et al (Lieb et al, 2018) overexpressed a *C. elegans* glutamate-gated Cl⁻ channel to establish an auto-regulatory mechanism, whereas both transgenes in our approach encode for endogenous proteins. More recently, Qiu et al. (Qiu et al, 2022) proposed a regulated system based on the expression of therapeutic genes under the control of an activity-dependent promoter (that is, a regulation at the transcriptional level), whereas our approach entails an autoregulation in which transgenes are continuously expressed, and the auto-regulatory loop is activated only in case of hyperactivity.

# Results

## NPY and Y2R overexpression in rat primary hippocampal neurons

To obtain cell-specific expression of NPY and Y2R, we designed a vector able to bias transgene expression towards excitatory neurons. Hence, transgenes were expressed under the control of the minimal CamKIIa(0.4) (mCamKIIa) promoter (Dittgen et al, 2004a), as shown in Fig. 1 (LV_NPY for NPY alone, LV_Y2-NPY for NPY and Y2R, and LV_EGFP (enhanced green fluorescent protein) as control). In LV_Y2-NPY, the Y2R gene was flag-tagged and separated from the NPY gene by the T2A self-cleaving peptide sequence. Both LV_NPY and LV_Y2-NPY also expressed EGFP, inserted downstream of either the NPY or the Y2R-NPY genes, and separated by an internal ribosome entry site (IRES) sequence.

We first tested the vectors in primary cultures of rat hippocampal neurons, to measure their ability to express NPY and Y2R by immunocytochemistry and western blot. As shown in Fig. 2A–C, the number of NPY-positive cells is increased in neurons transduced with either LV_NPY (Fig. 2B) or LV_Y2-NPY (Fig. 2C), compared to those transduced with LV_EGFP (Fig. 2A) in which only the subpopulation of endogenously expressing NPY interneurons is labeled. Western blot analysis on protein extracts confirmed that NPY levels are significantly increased in both LV_NPY and LV_Y2-NPY transduced cells (Fig. 2D,E), while Y2R expression, detected with the FLAG antibody, is present only in LV_Y2-NPY transduced cells (Y2-Flag, Fig. 2D). Incidentally, we also observed lower levels of EGFP in cultures treated with the "therapeutic" vectors than in those treated with the control vector (Fig. 2D); this is very likely due to the fact that IRES-driven gene translation is less efficient than the one driven directly by the promoter (Mizuguchi et al, 2000b).

To test the safety of this LV platform, we performed electrophysiological experiments on transduced primary hippocampal neurons and measured if overexpression of NPY-Y2, as compared with NPY alone, might influence normal neuronal activity. As shown in Fig. EV1, no significant difference was

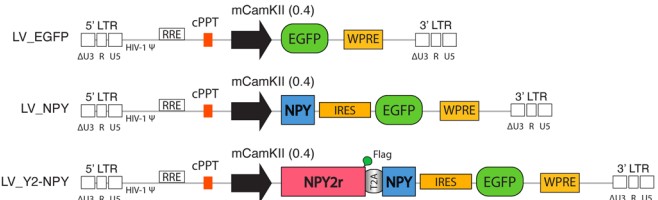

**Figure 1. Schematic representation of the LV design.**

Schematic representation of the lentiviral DNA (provirus), containing the EGFP (LV_EGFP), NPY (LV_NPY), or NPY and Y2R (LV_Y2-NPY) under the control of the mCamKIIa(0.4) promoter. In the LV_Y2-NPY construct, the Y2R gene is flag-tagged and separated from the NPY gene by the T2A self-cleaving peptide sequence. In the LV_NPY and LV_Y2-NPY constructs, EGFP expression is driven by an internal ribosome entry site (IRES) sequence. The scheme also represents the Woodchuck hepatitis virus regulatory element (WPRE); central polypurine tract (cPPT); rev responsive element (RRE); HIV-1 Ψ packaging sequence; and the long terminal repeats (LTR) that are self-inactivating (Zufferey et al, 1998) and composed by ΔU3, R, and U5.

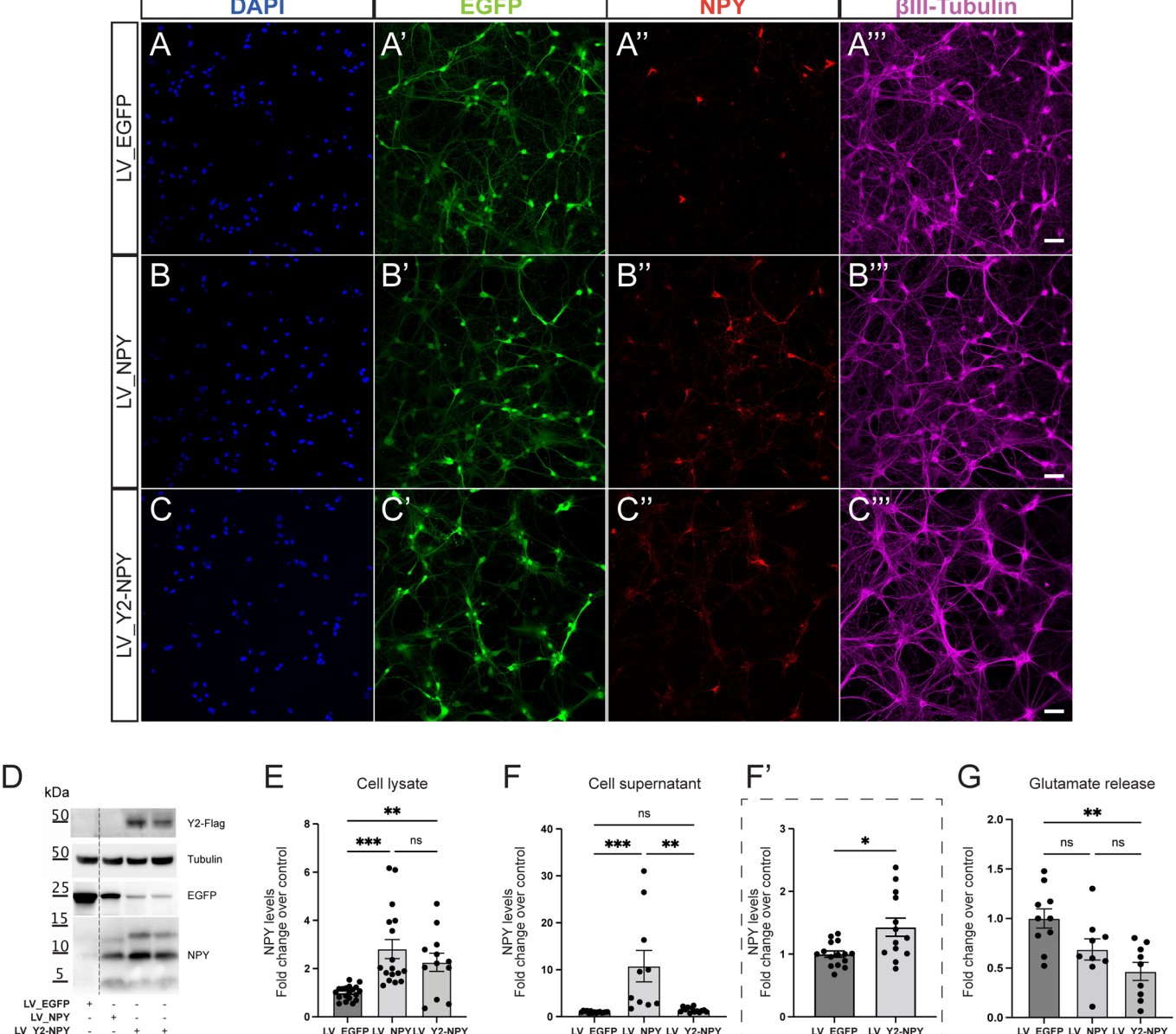

**Figure 2. Characterization of LV transduction in primary hippocampal neurons.**

(**A–C**) Representative confocal images of rat primary hippocampal neurons transduced with either LV_EGFP (**A-A'''**), LV_ NPY (**B-B'''**), or LV_Y2-NPY (**C-C'''**). Immunofluorescence staining for DAPI (blue), EGFP (green), NPY (red), and βIII-tubulin (magenta). Scale bar = 20 μm. (**D, E**) Representative western blot and quantification of the indicated proteins from total cell extracts of rat primary hippocampal neurons transduced with the specified vector. Protein signals were quantified and then normalized for loading (Tubulin). Results in (**E**) are the mean ± SEM of at least three independent experiments (biological replicates), with protein levels shown as fold change over control (LV_EGFP = 1 ± 0.06; LV_ NPY = 2.81 ± 0.39; LV_Y2-NPY = 2.26 ± 0.37; $p < 0.0001$). (**F, F'**). ELISA measurement of NPY extracellular levels in cell supernatants. Results are presented as mean ± SEM of at least three independent experiments (biological replicates), with peptide levels shown as fold change over control (LV_EGFP = 1 ± 0.05; LV_ NPY = 10.75 ± 3.34; LV_ Y2-NPY = 1.43 ± 0.14; (**F**) $p < 0.0001$; (**F'**) $p = 0.0143$). (**G**) Measurement of extracellular glutamate in transduced primary hippocampal neurons after 20 min of $Mg^{2+}$ withdrawal. Values were normalized for total protein content and are presented as means ± SEM of at least three independent experiments (biological replicates), with glutamate levels shown as fold change over control (LV_EGFP = 1.00 ± 0.10; LV_ NPY = 0.69 ± 0.11; LV_ Y2-NPY = 0.47 ± 0.09; $p = 0.0056$). Statistical significance was calculated using the Kruskal–Wallis one-way analysis of variance followed by Dunn's post hoc test (**E–G**) or by the Mann–Whitney test for (**F'**) (note that in **F'** are shown data comparing only values for LV_EGFP and LV_Y2-NPY from panel **F**). *$p < 0.05$; **$p < 0.01$; ***$p < 0.0001$. Source data are available online for this figure.

observed in the basic electrophysiological properties of neurons transduced with either LV_ NPY or LV_Y2-NPY.

Finally, we performed an ELISA assay to measure NPY release by transduced neurons. As expected, cells transduced with LV_NPY displayed the highest levels of NPY in the supernatant (Fig. 2F). NPY extracellular levels were also increased by LV_Y2-NPY (Fig. 2F'), but less effectively than by LV_NPY. In addition, we measured the amount of glutamate released by transduced

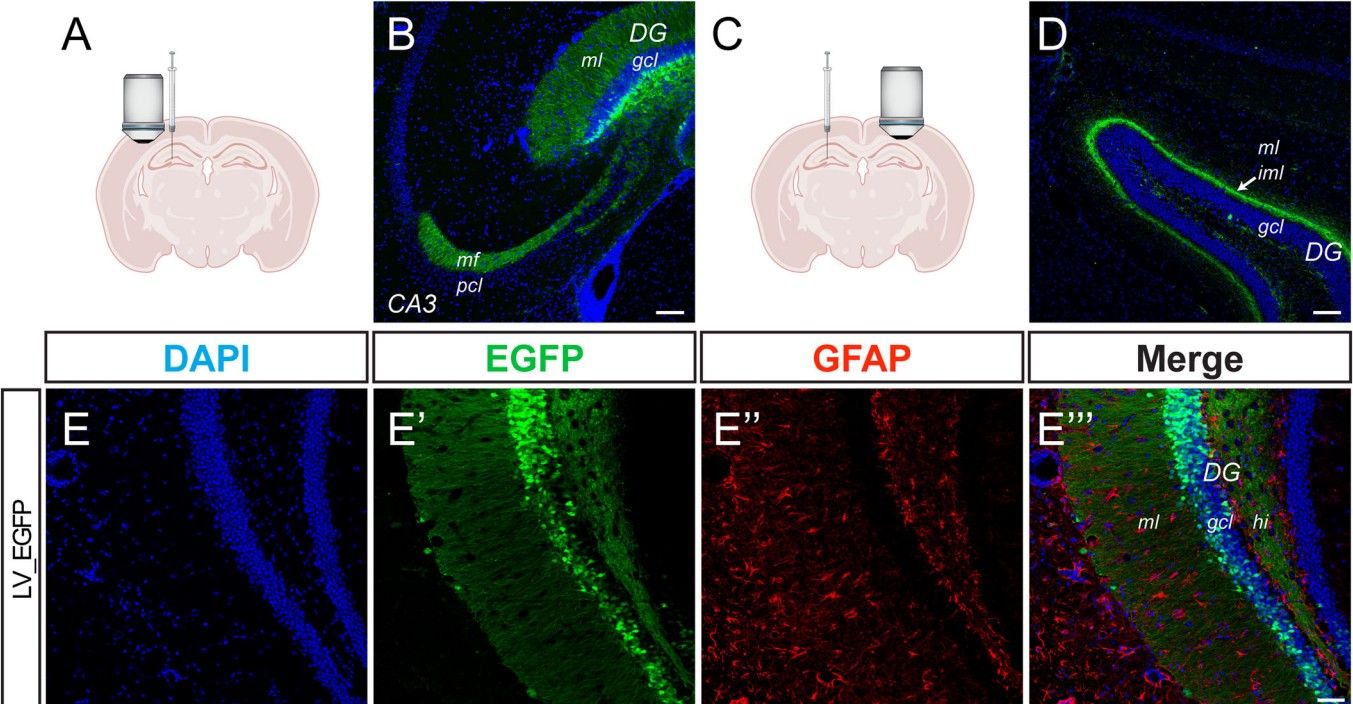

**Figure 3. Characterization of LV_EGFP expression in the DG of the WT mouse.**

(A) Schematic representation of the mouse brain showing the site of viral vector injection and the field imaged in (B). (B) Representative image of the hippocampus ipsilateral to the injection of LV_EGFP immunostained for DAPI (blue) and EGFP (green). Scale bar = 100 µm. (C) Schematic representation of the mouse brain showing the site of viral vector injection and the field imaged in (D). (D) Representative images of the hippocampus contralateral to the injection of LV_EGFP immunostained for DAPI (blue), and EGFP (green). Scale bar = 100 µm. (E) Representative GFAP staining of the dentate gyrus in the ipsilateral hippocampus. Staining for DAPI (blue), EGFP (green), GFAP (red), are shown. Scale bar = 50 µm. DG dentate gyrus, ml molecular layer, gcl granular cell layer, CA3 Cornu Ammonis 3 mf, mossy fiber pcl pyramidal cell layer, iml inner molecular layer (indicated by an arrow in panel D), hi hilus. Source data are available online for this figure.

neurons under conditions in which neuronal activity was increased by the elimination of $Mg^{2+}$ from the medium (0 $Mg^{2+}$) (Hogg et al, 2019; Mele et al, 2021) (Fig. EV2). As shown in Fig. 2G, although the difference between the effect of LV_NPY and LV_Y2-NPY was not statistically significant, glutamate release was significantly reduced only in cultures transduced with LV_Y2-NPY. Taken together, these data suggest the expression of the Y2R can establish an autofeedback on the release of both glutamate and NPY.

## NPY and Y2R overexpression in excitatory neurons of the DG in the mouse hippocampus

To test our viral vector platform in vivo, we first injected wild-type (WT) mice with a single dose of control vector (LV_EGFP, 2 µl of $1 \times 10^9$ TU/mL viral preparation, i.e., $2 \times 10^6$ TU total) in the brain parenchyma (coordinates −1.8 AP; ±1.8 ML; −2.0 DV) targeting the DG of the dorsal hippocampus (Fig. 3A). LV transduction was achieved for the most part in GCs, as indicated by the positivity of cell bodies in the granular layer of the DG, of the DG molecular layer (presumably due to GC dendrites), and of nerve terminals in the polymorphic layer and in the stratum lucidum of CA3 (Fig. 3B). We also detected EGFP-labeling of the contralateral inner molecular layer, indicating transduction of ipsilateral mossy cells projecting commissural fibers (Fig. 3C,D).

To evaluate the cell sub-type specificity of the mCamKIIa promoter, a subset of animals was used to assess EGFP expression patterns in hippocampal slices, co-stained with different cellular markers. Consistent with the previous characterization of the promoter activity in cortical cells (Dittgen et al, 2004a), no signal was found in hippocampal glial fibrillary acidic protein (GFAP)-positive astrocytes (Fig. 3E). In addition, except for expression in a minor subset of CA3 (but not CA1 and DG) GAD-1 positive (Fig. 4A,E) and parvalbumin (PV)-positive interneurons (Fig. 4B,F), EGFP expression was negligible in somatostatin (SOM)- and in NPY-positive GABAergic neurons (Fig. 4C,D,G,H).

We then focused our attention on the overexpression of NPY and Y2R. WT mice were injected with a single dose of LV_NPY or LV_Y2-NPY in the dorsal hippocampus (dose and coordinates as above), and expression of NPY and Y2R was assessed by immunofluorescence and western blot. Coherently with in vitro experiments, strong over-expression of NPY was detected in transduced neurons. In particular, mossy fiber terminals were strongly immunoreactive for NPY staining in animals that received either LV_NPY or LV_Y2-NPY, but not in those receiving LV_EGFP (Fig. 5A–C). Similarly, Y2R immune-positive fibers were detected in LV_Y2-NPY transduced animals, with a pattern resembling the one previously described (Stanić et al, 2011), but not in those receiving LV_NPY or LV_EGFP (Fig. 5D,E). Although endogenous Y2R expression has been described on

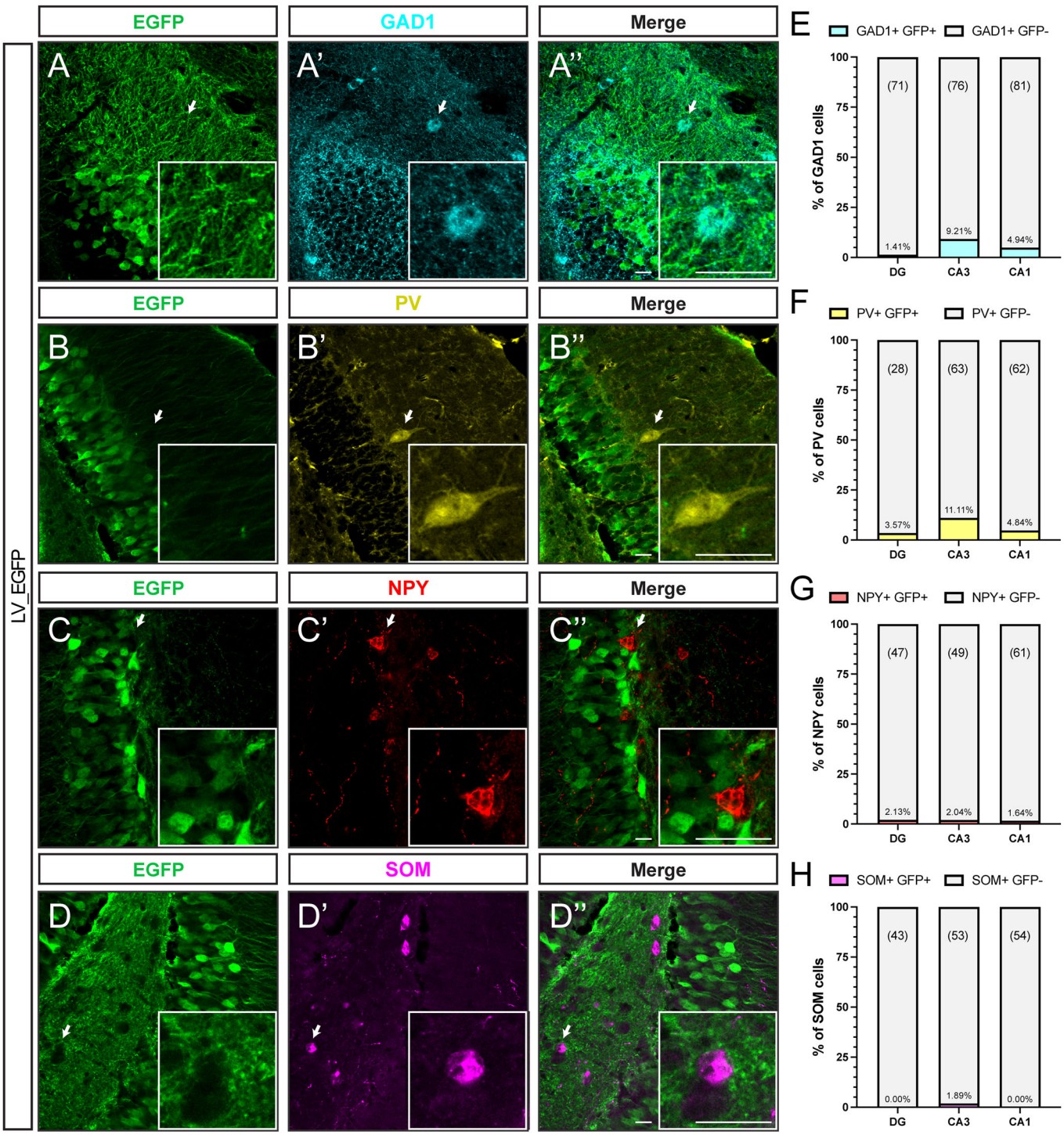

**Figure 4. mCamKII(0.4) promoter-driven expression of EGFP in hippocampal interneurons.**

(A–A″, B–B″, C–C″, D–D″) Representative confocal images of the dentate gyrus (DG) of fluorescently immunolabeled glutamate decarboxylase 1 (GAD-1, cyan), parvalbumin (PV; yellow), neuropeptide Y (NPY; red), somatostatin (SST; magenta) containing interneurons, together with *EGFP* (*green*) expression of animals transduced with LV-EGFP vector. Arrows indicate interneurons labeled with various markers and lacking EGFP signal. Scale bars = 20 μm. (E–H) Percentage of EGFP-expressing cells of the different interneuron sub-types in DG, CA3, and CA1 regions of the hippocampus. N = 3 mice. Number of examined cells in a given region is indicated above each bar. Source data are available online for this figure.

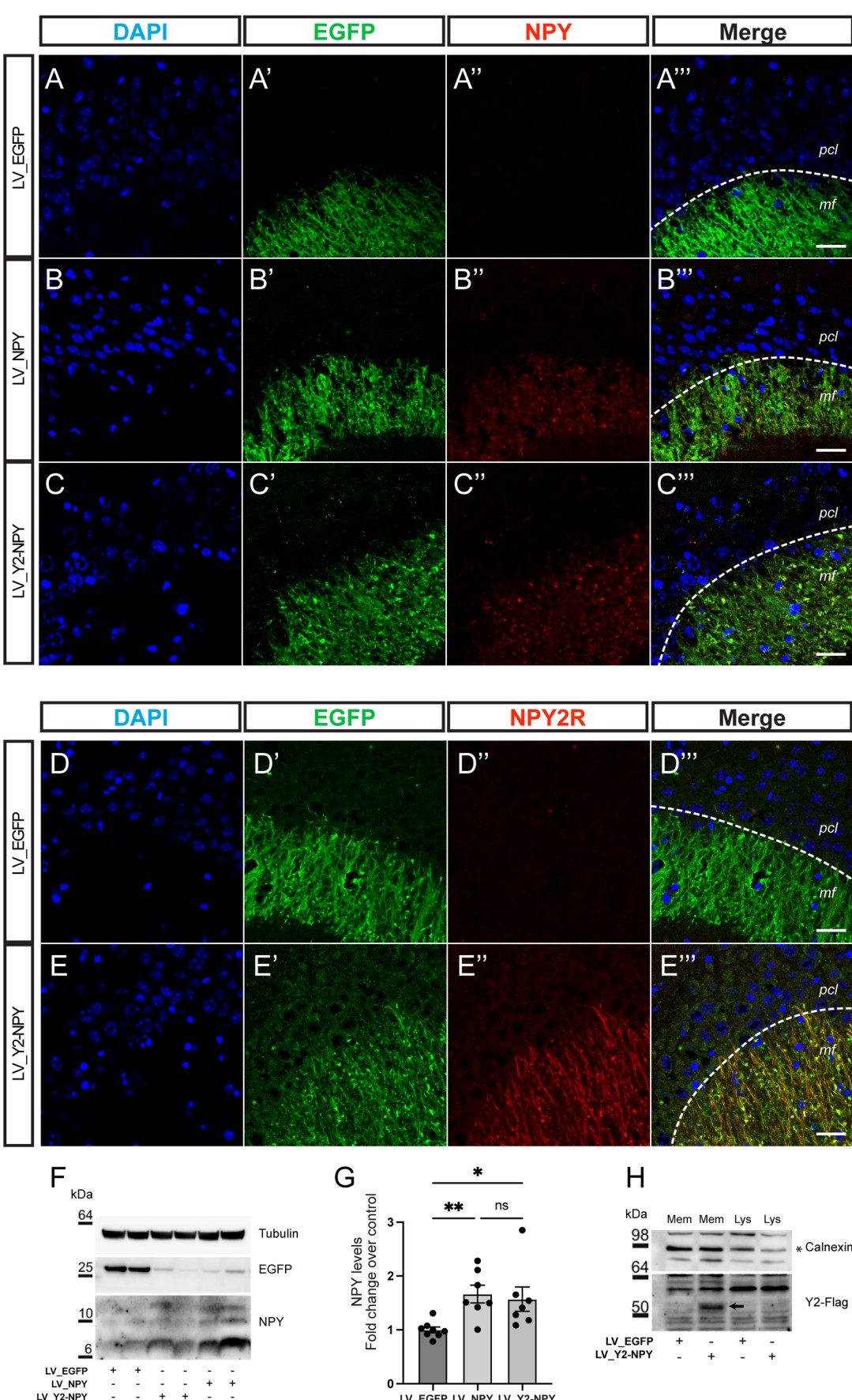

◄ **Figure 5. Characterization of LV transgene expression in the WT mouse.**

(A–E) Representative confocal images of immunofluorescence performed on the WT mouse brain injected with either LV_EGFP (A-A''' and D-D'''), LV_NPY (B-B'''), or LV_Y2-NPY (C-C''' and E-E'''). The green signal highlights the mossy fiber pathway of transduced dentate granule cells projecting their axons onto CA3 pyramidal neurons. (A–C) Staining for DAPI (blue), EGFP (green), and NPY (red) are shown. (D, E) Staining for DAPI (blue), EGFP (green), Y2R (red), are shown. Scale bar = 20 μm. (F, G) Representative western blot (F) and quantification (G) of the indicated proteins in extracts from hippocampi of WT mice injected with the indicated vectors. Protein levels are normalized for loading (Tubulin) and are shown as fold change over control (LV_ EGFP = 1 ± 0.05, $n$ = 8 mice; LV_NPY = 1.66 ± 0.16, $n$ = 7 mice; LV_Y2-NPY = 1.57 ± 0.22, $n$ = 7 mice $p$ = 0.0007). Values are presented as means ± SEM. Statistical significance was calculated using the Kruskal–Wallis one-way analysis of variance followed by Dunn's post hoc test. *$p$ < 0.05; **$p$ < 0.01. (H) Representative western blot of the indicated proteins in the membrane fraction (Mem) or total extract (Lys) of hippocampi from WT mice injected with the indicated vectors. The arrow indicates the Y2-flag band in the western blot image. pcl pyramidal cell layer, mf mossy fiber. Source data are available online for this figure.

presynaptic terminals of mossy fibers in rat, mouse, and human samples (Stanić et al, 2006, 2011; Furtinger et al, 2001; Sperk et al, 1992), we were not able to detect its expression in animals untreated or treated with LV_EGFP (Fig. 5D).

While we did not monitor the animals' health by using specific tests after direct injection of the LV_ NPY or LV_Y2-NPY vectors in the hippocampus, we did not observe any change in their gross behavior (including aggressiveness and feeding behavior), weight curve and EEG patterns (including that during sleep), confirming the above-described in vitro experiments on the safety of these vectors.

Consistent with these results, western blot analysis of injected animals confirmed a significant overexpression of NPY and Y2R by the "therapeutic" vectors. We found a higher level of NPY expression in hippocampal extracts from animals treated with LV_NPY or LV_Y2-NPY, as compared with LV_EGFP (Fig. 5F,G). Since Y2R expression was difficult to detect in tissue extracts, we prepared hippocampal membrane fractions from injected animals that, as expected, displayed enrichment for Y2R only in LV_Y2-NPY transduced tissue (Fig. 5H).

## Upregulation of NPY in the mossy fiber terminals during epileptogenesis

NPY levels are increased in epileptic hippocampal samples from animal models and patients with focal epilepsy (Furtinger et al, 2001; Sperk et al, 1992). Data in rodent models of mesial temporal lobe epilepsy suggest that it is due to ectopic expression in excitatory neurons not normally expressing it (in particular, GCs) (Mathern et al, 1995; McCarthy et al, 1998), because GABAergic interneurons that physiologically express NPY are instead lost (Huusko et al, 2015). We evaluated the regulation of NPY expression in the TKO mouse model, in which deletion of the three synapsin genes results in spontaneous seizures. The TKO mouse model was selected because it meets some fundamental prerequisites: (1) the model exhibits a "balanced" epileptic phenotype, i.e., not too mild, and not too severe; (2) the time course of spontaneous seizure appearance in the life of the animal is well defined; (3) the mechanism of seizure generation is reasonably well known, and representative of general seizure generation mechanisms already hypothesized or identified in other models. In fact, as previously described (Cambiaghi et al, 2013), TKO animals do not display any gross defect nor epileptic phenotype during early development. However, increased susceptibility to seizures and overt (i.e., clearly detectable) spontaneous seizures, which are not too infrequent for proper video-EEG monitoring and are not fatal (point 1 above) are observed between

60 and 100 days of life, with a peak of frequency between 60 and 90 days (point 2 above). Furthermore (point 3), a significant reduction in the hippocampal release of the inhibitory neurotransmitter gamma-aminobutyric acid (GABA) from GABAergic interneurons (to a greater extent than that of glutamate from glutamatergic neurons) has been shown to occur in TKO hippocampi (Cambiaghi et al, 2013; Farisello et al, 2013). Impairment of GABA signaling is thought to play a causal role in seizure generation in this model, being a common mechanism of seizure generation (Shimoda et al, 2023).

We first asked if NPY ectopic expression in DG GCs might represent a compensatory mechanism to seizure generation. To this aim, we evaluated if TKO animals display adaptative alterations in the NPY expression in a period ranging between 30 to 120 days of life. We first performed a temporal assessment of NPY expression in the hippocampi of WT and TKO mice at different timepoints: 30, 60, 90, and 120 postnatal days (p30, p60, p90, and p120). Interestingly, NPY expression levels in TKO animals increased dramatically at p90, after the period in which most of the animals were experiencing the highest seizure activity, and then returned to baseline levels at a later timepoint (p120 - Fig. 6A,B), when spontaneous seizures were more rare. We then performed an immunohistochemistry analysis for NPY on WT and TKO animals at 100 days of life. As shown in Fig. 6C,D, we observed an ectopic NPY expression in the mossy fiber terminals of TKO mice, which is not present in age-matched WT animals.

## Combined overexpression of NPY and Y2R reduces the frequency and duration of seizures in TKO mice

One possible explanation of the increased levels of NPY at p90-p100 can be that it represents an adaptive reaction to spontaneous seizures, that (as per Cambiaghi et al, 2013) we observed to occur at the highest frequency between p60 and p90. In fact, findings in other models (Vezzani and Sperk, 2004; Cattaneo et al, 2021) support the notion that reactive NPY overexpression follows intense seizure activity. Ectopic overexpression of NPY in GCs may be hypothesized to quench hyperexcitability in this key cell population and thereby reduce seizure occurrence. If this were the case, anticipating and reinforcing this NPY response should prevent seizure occurrence. Therefore, we decided to treat TKO mice in a time window when seizures are present, but the ectopic expression of NPY is still absent, or at least undetectable.

As shown in the schematic representation of Fig. 7A, we treated TKO littermates with bilateral injections of LV_EGFP or LV_Y2-NPY in both hippocampi at p45 and monitored seizure development for 3 consecutive weeks, beginning at p60 to allow

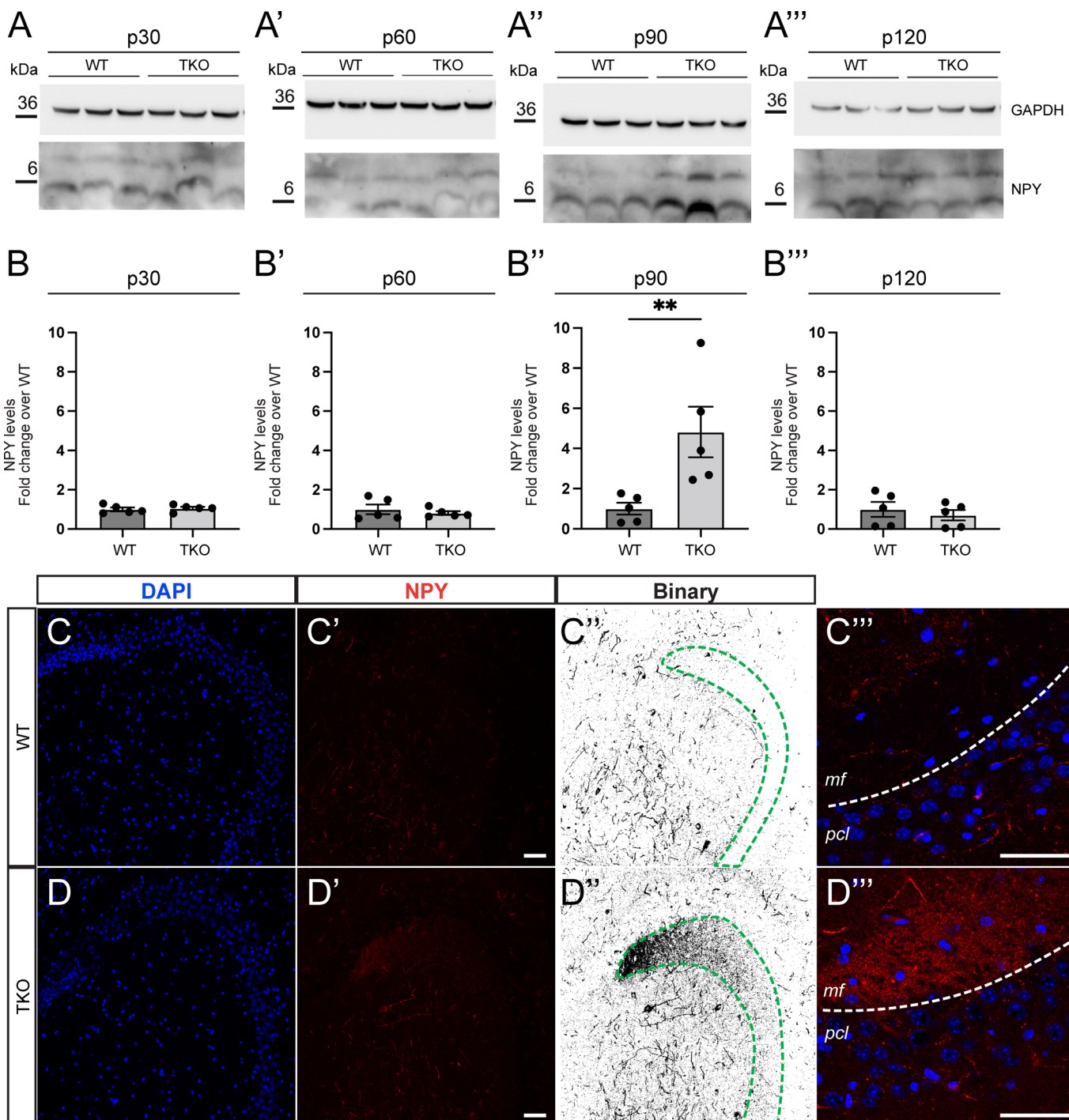

**Figure 6. Upregulation of NPY in the hippocampus of the adult TKO mouse.**

(A, B) Representative western blot (A-A''') and quantification (B-B''') of NPY in extracts from hippocampi of TKO mice at the indicated timepoints: 30 postnatal days (p30) for A and B; 60 postnatal days (p60) for A' and B'; 90 postnatal days (p90) for A'' and B'' and 120 postnatal days (p120) for A''' and B'''. Protein levels were normalized for loading (GAPDH) and are shown as fold change over control (WT). Data were the mean ± SEM of five mice for each genotype for each timepoint. (B) WT = 1 ± 0.09; TKO = 1.05 ± 0.07; $p = 0.8413$ (B') WT = 1 ± 0.24; TKO = 0.08 ± 0.09; $p > 0.9999$ (B'') WT = 1 ± 0.29; TKO = 4.82 ± 1.26; $p = 0.0079$ (B''') WT = 1 ± 0.37; TKO = 0.7 ± 0.26. $p = 0.4206$; Statistical significance was calculated using the Mann–Whitney $U$-test. **$p < 0.01$. (C, D) Representative confocal images of WT (C-C''') and TKO (D-D''') mice at 100 days of age. Immunofluorescence staining for DAPI (blue) and NPY (red and binary) is shown. Note the mossy fiber pathway, containing dentate granules projecting axons onto CA3 pyramidal neurons, which is ectopically expressing NPY only in the TKO mice. Scale bar = 50 µm. C''' and D''' show a higher magnification of the mossy fiber at the CA3 level of WT (C) and TKO (D) mice. Immunofluorescence staining for DAPI (blue) and NPY (red). Scale bar = 50 µm. pcl pyramidal cell layer, mf mossy fiber. Source data are available online for this figure.

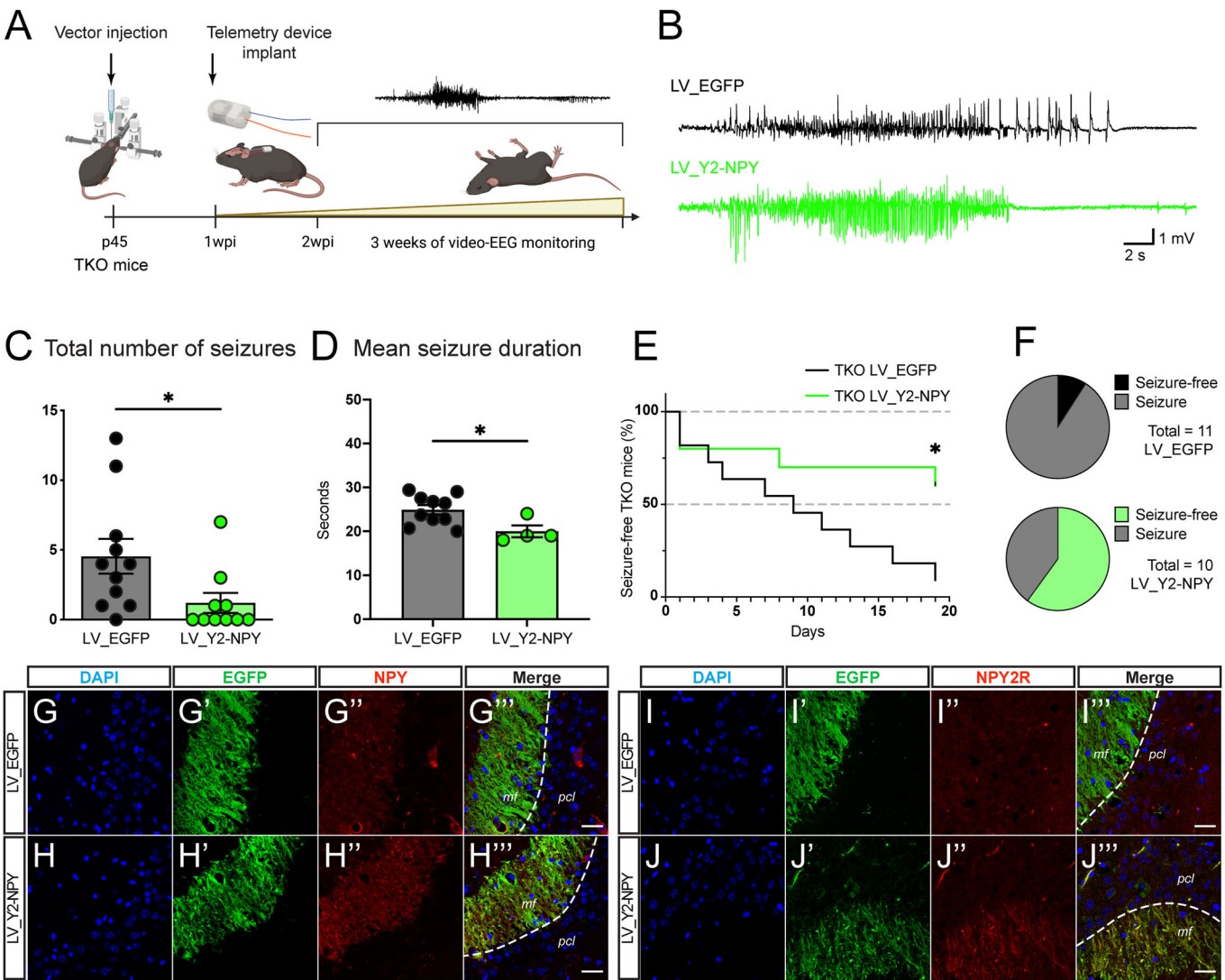

**Figure 7. LV overexpression of NPY and Y2R reduces seizures in the TKO mouse.**

(A) Schematic illustration of the experimental set-up for the treatment of TKO mice. Viral vector injection of either LV_EGFP or LV_Y2-NPY was performed at p45. EEG-telemetry electrode was implanted 1 week post viral vector injection. Video-EEG monitoring started 2 weeks after viral vector injection (p60). (B) Representative EEG traces of behavioral seizures in LV_EGFP and LV_Y2-NPY treated TKO mice. (C) Mean number of seizures for each group. Dots represent the total number of seizures for each animal. Data were shown as means ± SEM; LV_EGFP = 4.5 ± 1.2, and LV_Y2-NPY = 1.2 ± 0.7. (D) Mean seizure duration for each group. Dots represent the mean seizure duration for each animal. Note that only animals that displayed seizures were included in the LV_Y2-NPY group. Data were shown as means ± SEM; LV_EGFP = 24.9 ± 1 s, and LV_Y2-NPY = 20 ± 1.3 s. *$p < 0.05$; Mann–Whitney U-test for (C) ($p = 0.0147$) and (D) ($p = 0.0120$). (E) Kaplan–Meier plot showing the time to first seizure from the start of monitoring for each group ($n = 11$ TKO mice treated with LV_EGFP, $n = 10$ TKO mice treated with LV_Y2-NPY). *$p = 0.014$; Log-rank (Mantel-cox) test. (F) Pie charts showing the percentage of LV_EGFP and LV_Y2-NPY treated TKO mice with or without spontaneous seizures at the end of the experiment. $p = 0.0237$; Fisher's exact test. (G–J) Representative confocal images of TKO mice following video-EEG monitoring treated with either LV_EGFP (G-G''' and I-I''') or LV_Y2-NPY (H-H''' and J-J'''). Immunofluorescence staining for DAPI (blue), EGFP (green), and NPY (red) are shown in G-H. Immunofluorescence staining for DAPI (blue), EGFP (green), and Y2R (red) are shown in (I, J). Scale bar = 20 μm. pcl pyramidal cell layer, mf mossy fiber. Source data are available online for this figure.

2 weeks of recovery from injection and achieve a stable transgene expression, and 1 week of recovery from electrode implantation. To detect EEG and motor seizures, we employed a telemetry system that allows continuous video-EEG monitoring, 24/24h for 7 days a week (Fig. 7B).

The combined expression of NPY and Y2R in LV_Y2-NPY treated animals significantly reduced the total number of seizures observed during these 3 weeks of continuous video-EEG recording (Fig. 7C). In these animals, the duration of the few occurred

individual seizures was also significantly reduced (Fig. 7D). It is not possible to accurately measure the time to first seizure in our experimental settings, because full expression of transgenes takes a few days and monitoring was performed starting 2 weeks after vector injection. However, animals treated with LV_Y2-NPY displayed a delayed appearance of seizures from the start of video-EEG recording ($p = 0.0149$, log-rank Mantel-Cox test; Fig. 7E), and the percentage of TKO mice that displayed at least one seizure by the end of the monitoring procedure was

significantly reduced in the LV_Y2-NPY compared with the LV_GFP group ($p = 0.0237$; Fisher's exact test; Fig. 7F).

Animals were sacrificed at the end of the monitoring period to confirm the correct expression of the transgenes in the TKO background. In line with what was previously reported in other non-genetic models (Sperk et al, 1992), an increased expression of NPY was observed in TKO mice treated with LV_EGFP in response to seizure occurrence (Fig. 7G). This was further increased by the injection of LV_Y2-NPY (Fig. 7H). Moreover, no Y2R positive fibers were detectable in control (LV_EGFP) treated TKO animals (Fig. 7I), while Y2R positive terminals were observed in the mossy fibers terminals in animals treated with LV_Y2-NPY (Fig. 7J).

# Discussion

In the present work, we show that overexpression of NPY and its Y2R in hippocampal excitatory neurons (in our settings, predominantly in DG GCs) exerts robust anti-seizure effects in the Synapsin triple-KO model of epilepsy. Overexpression in excitatory neurons was achieved through the use of a single LV expressing both Y2R and NPY under a mCamKIIa promoter. This vector design is expected to lead to hetero-control: NPY released by transduced cells will modulate the activity of synaptically connected neurons carrying NPY receptors, and NPY-releasing neurons will inhibit synaptically connected Y2R-transduced cells. However, the design is primarily expected to lead to auto-control of NPY- Y2R transduced neurons that, whenever hyperactive, would release NPY, which in turn would dampen excitation via overexpressed Y2R. Here, we provide initial evidence of the establishment of this auto-inhibitory loop. First, since both transgenes are expressed by each single viral particle, they must be expressed together in each transduced cell and, in fact, we observed localization of both NPY and Y2R in mossy fiber terminals, that is, the anatomical substrate of autoregulation. In addition, we observed that NPY extracellular levels in vitro are increased much less by the vector expressing NPY and Y2R than by the vector expressing NPY alone, and that glutamate release is significantly reduced by the vector expressing NPY and Y2R but not by the vector expressing NPY alone. The most parsimonious interpretation of these data is the existence of an autofeedback inhibition on a glutamatergic terminal.

Because hyperactivity of DG GCs is thought to be critical for the development and maintenance of a temporal lobe seizure network (Krook-Magnuson et al, 2015), we hypothesized that this auto-inhibitory loop could control seizures of different etiologies involving the hippocampus. In fact, it has been hypothesized that a subpopulation of highly connected GCs (the superhubs) are key drivers of epileptiform activity and that closed-loop interventions targeting these cells can control epileptic networks (Hadjiabadi et al, 2021). If this hypothesis is correct, seizure prevention would require the control of a relatively small group of neurons and not of multiple cell populations nor even of the majority of GCs.

Indeed, we provide here evidence that a biased expression of Y2R and NPY in hippocampal GCs is sufficient to exert a significant seizure-suppressant effect in a genetic model of epilepsy, the TKO. Although several gene therapy approaches based on NPY and Y2R, have been investigated in pre-clinical models of acquired epilepsy (Noè et al, 2008, 2010; Nikitidou Ledri et al, 2016;

Szczygieł et al, 2020b), so far, only one study has been conducted in a genetic model, specifically a model of absence seizures (Powell et al, 2018) in which NPY, delivered in the thalamus and somatosensory cortex via AAV-mediated transduction, proved effective in suppressing absence-like seizures. It should be emphasized that we did not expect our approach to be superior to previous ones in terms of seizure control. The above-cited previous studies overexpressed NPY and/or Y2 in a non-cell-specific, non-regulated manner, ensuring a constant and widespread increase of this inhibitory signal. We instead aimed at testing if restricting expression to excitatory neurons (for what possible, to hippocampal GCs) in an auto-regulated manner was sufficient to produce effects on seizures.

# Vector design

In our vector design, the two therapeutic genes are separated by T2A, avoiding the use of an IRES. IRES is known to reduce the expression level of the downstream gene and limit the processing of sequential genes in the same multi-cistronic expression cassette, particularly for gene products that are delivered to different subcellular compartments (Melin et al, 2023; Mizuguchi et al, 2000a; Hennecke, 2001; Bochkov and Palmenberg, 2006). We took instead advantage of the T2A "self-cleaving" system, which allows a higher-level expression of downstream genes in multi-gene cassettes (Liu et al, 2017; Hadpech et al, 2018). During translation, the presence of the T2A sequence leads to the inclusion of additional amino acids upstream and downstream of the genes of interest, which may impact the function of the final protein product. However, NPY, the second gene in our design, is translated as a pre-propeptide that is physiologically processed through sequential cleavages at both the N- and C-terminal region (Cattaneo et al, 2021), leading to the formation of a propeptide and then of the mature form. In our settings, therefore, both the pro- and the mature peptides were identical to their physiological counterparts (as shown in Fig. 2D).

We found that, whereas the increase of NPY intracellular levels was similar with LV_NPY and LV_Y2-NPY in neuronal cultures, a higher increase in the extracellular levels of NPY was obtained with the LV_NPY vector. This observation is expected based on the establishment of the auto-inhibition loop described above, because NPY release in neurons transduced with LV_Y2-NPY, by activation of Y2R in the very same neurons, should result in a reduction of NPY release itself.

The mCamKIIa promoter proved sufficient to bias transgene expression in hippocampal excitatory neurons. In our study, we used the minimal CamKIIa promoter sequence described by Dittgen et al (Dittgen et al, 2004b), in a LV pseudotyped with the vesicular stomatitis virus (VSV) glycoprotein. VSV-G-pseudotyped LV proved efficient in transducing excitatory neurons in mice, rats, and monkeys (Yaguchi et al, 2013). By using the control vector LV_EGFP, in which EGFP is driven by mCamKIIa, we did not observe EGFP expression either in astrocytes (GFAP+), parvalbumin-positive interneurons (PV+), SOM, or NPY-expressing inhibitory interneurons. Our data were in line with the observation made by the above authors in cortical pyramidal neurons. Other studies employed a different vector platform, namely AAV. AAV2/5 and AAV2/9 displayed CamKII-driven

expression in five different subclasses of GABAergic interneurons in the mouse cortex (Veres et al, 2023), while AAV injection in the hippocampus showed negligible or no immunoreactivity for the reporter gene. On the same line, an AAV2/1 expressing the humanized Renilla reniformis GFP (hrGFP) under the CamKII promoter labeled PV interneurons in the marmoset cortex (Watakabe et al, 2015). A conservative interpretation of these results suggests that, when using an AAV vector for the expression of transgenes driven by the CamKII promoter, caution should be used in assuming selectivity in excitatory neurons, due to a potential preference of the vector for transducing inhibitory neurons, especially in cortical areas.

By using the control vector LV_EGFP, in which EGFP is driven by mCamKIIa at the highest expression level, we did not observe EGFP expression either in astrocytes (GFAP+), parvalbumin-positive interneurons (PV+), SOM or NPY-expressing inhibitory interneurons. In addition, we demonstrated that both LV_NPY and LV_Y2-NPY efficiently overexpress NPY in the mossy fiber terminals of the dentate gyrus, which are mainly composed of the axon terminals of excitatory dentate granule neurons. These findings are in line with a recent neuroanatomical and electrophysiological characterization of the expression pattern of the full sequence of the CamKIIa promoter (~1.2 kb) in the mouse hippocampus, prefrontal cortex, and basolateral amygdala (Veres et al, 2023) where only a few inhibitory neurons were labeled with the use of the CamKIIa promoter in the hippocampal region. Once again, these data support the notion that biasing overexpression of NPY and Y2R in hippocampal excitatory cells, by boosting the NPY-Y2R auto-inhibitory axis only in excitatory cells, reducing glutamate release, and avoiding the establishment of the same feedback loop in the GABAergic population, can ensure a robust anti-seizure effect.

## Anti-seizure effects

In this study, we employed a genetic model of epilepsy, the TKO mouse. As mentioned in the Results section, this model (1) presents a "balanced" epileptic phenotype; (2) displays a well-defined time course of seizure appearance in the life of the animal, offering the possibility to identify a therapeutic window during which seizures are not yet occurring but the epileptogenic process is ongoing; (3) underlies a putative mechanism of seizure generation, which is common to most epilepsy models, i.e., impairment of GABA signaling.

In addition, we found that the TKO model reproduces the typical features of the pathophysiological alterations in the NPY system that have been extensively described in both rodent and human epileptic samples (Sperk et al, 1992; Furtinger et al, 2001). In the epileptic hippocampus, NPY is ectopically expressed by DG GCs, while its expression by GABAergic interneurons is reduced (Noe' et al, 2007; Sørensen et al, 2008). We found that similar adaptive mechanisms take place in the TKO model. It was previously shown that motor seizures in the TKO model begin to occur at around one month of life and reach a peak at approximately 90 days of age, when 90% of the animals had experienced at least one behavioral seizure, before declining in the following months (Cambiaghi et al, 2013). We hypothesized that NPY upregulation might be instrumental in this reduction in

seizure frequency. Hence, we performed a spatiotemporal analysis of NPY upregulation with quantitative measurements at four key timepoints (30, 60, 90, and 120 days of age). In keeping with the hypothesis, we observed an increase in the levels of NPY in the hippocampus of TKO mice at 90 days of age, following the period of highest seizure frequency, with expression localized mainly ectopically, in DG GCs, like in other epilepsy models and in humans with mesial temporal lobe epilepsy.

The anti-correlation between seizure occurrence and NPY levels has been known for decades (Bellmann et al, 1991). In the pathophysiological settings, however, the efficacy of NPY (in particular, its ectopic expression in the dentate gyrus) as an endogenous anticonvulsant may be limited in time. For example, in preconditioned animals in which sub-threshold kainic acid was used to induce NPY upregulation, the neuropeptide was able to reduce pyramidal cell loss only within a limited time window (1–7 days but not at 15 days (El Bahh et al, 2001)). Recently, in a model based on selective inhibition of GABA release from PV terminals in the subiculum, it has been shown that NPY positivity in interneurons persists for several weeks, whereas NPY levels in the DG GCs decreases between 5 and 7 days after the last seizure (Drexel and Sperk, 2022). Hence, NPY persistence in GABAergic cells and not in GCs may be responsible for the reduction in spontaneous recurrent seizures observed in this model.

We treated TKO animals with LV_Y2-NPY before the occurrence of the endogenous NPY upregulation, with the aim of anticipating this adaptive mechanism. The expectation was to prevent the onset or attenuate the severity of spontaneous seizures. Existing data (Krook-Magnuson et al, 2015) support the notion (in keeping with the so-called dentate "gate" hypothesis) that preventing GC hyperexcitability can prevent the occurrence of spontaneous seizures. Thus, an NPY-Y2R auto-regulatory loop, by avoiding excessive GC activity and glutamate release, should prevent seizures, that is, seizure frequency should be reduced whereas rare, occasional escape to control should evoke nearly "normal" seizures.

Indeed, we observed that the combination of NPY and Y2R reduced the total number and (to a lesser level) the mean duration of seizures. Moreover, it decreased the percentage of animals that display at least one seizure by the end of the monitoring window (3 weeks). Further studies will be needed to evaluate the long-term persistence of this effect. The reasons to start testing our system in the TKO model notwithstanding, it will also be needed in the future to extend the present observations to other epilepsy models. Nonetheless, these data demonstrate that overexpression of NPY and Y2R in DG GCs is sufficient to exert a robust anti-seizure effect in a genetic model of epilepsy.

## Concluding remarks

In this work, we developed and characterized an innovative NPY-based vector platform, targeting excitatory neurons in an auto-regulated manner. Data provide evidence that a gene therapy approach modulating a basic mechanism of seizure control (i.e., rebalancing excitation/inhibition) can be effective for treating a genetic form of epilepsy. Data also suggest that targeting a (sub) population of cells (DG GCs, maybe even a subset of GCs that are active drivers of epileptic activity (Hadjiabadi et al, 2021)) can be

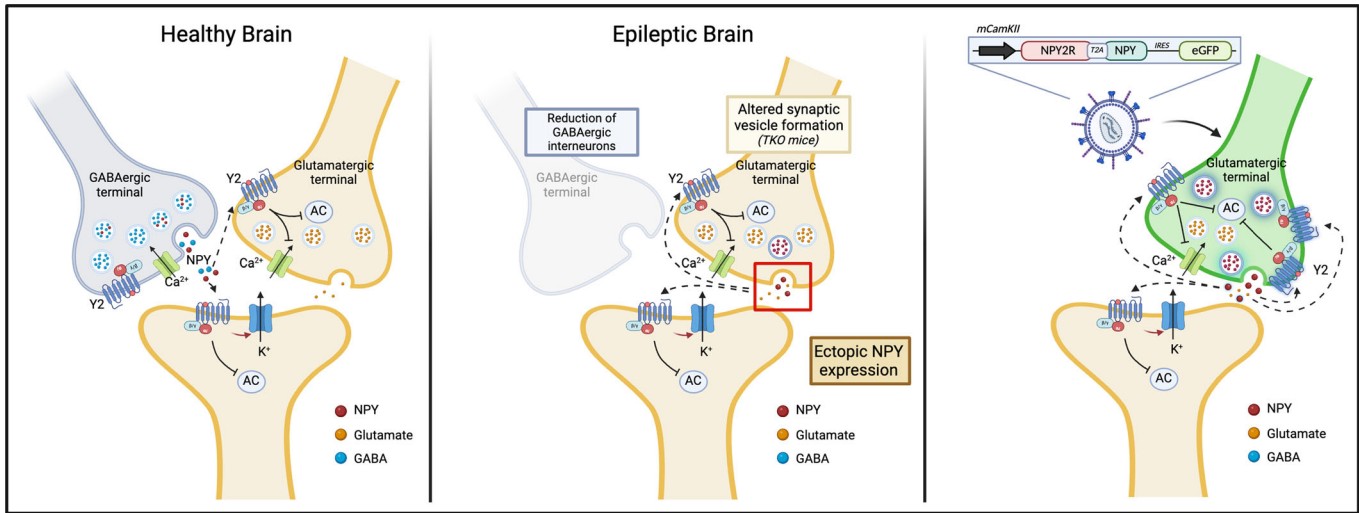

**Figure 8. Proposed auto-regulated mechanism of the anti-seizure effect of LV_Y2-NPY.**

Schematic representation containing three comparative diagrams of synapses in healthy and epileptic brains, along with a diagram illustrating the proposed intervention. The Healthy Brain shows a CA3 synapse with a GABAergic terminal on the left and a mossy fiber glutamatergic terminal on the right. A post-synaptic neuron is shown at the bottom. In the interneurons, GABA (blue dots) is contained in small synaptic vesicles and is also co-stored with NPY (red dots) in large dense core vesicles. The glutamatergic terminal releases glutamate (orange dots). Y2 receptors for NPY are present on both terminals. The epileptic brain is represented by a reduction of GABAergic interneurons and ectopic expression of NPY in glutamatergic terminals. The gene therapy intervention (right panel) consists of the overexpression of NPY2R and NPY into glutamatergic terminals, which modulate synaptic activity; the expression of Y2R by the same neurons that release NPY is expected to result in an auto-inhibitory feedback when neuronal activity is excessive.

sufficient to prevent spontaneous seizures (Fig. 8). Together, these results suggest that an auto-regulated gene therapy approach based on co-expression of NPY and Y2R in hippocampal GCs is feasible and worth further investigation.

## Methods

### Reagents and tools table

| Reagent/Resource | Reference or source | Identifier or catalog number |
|---|---|---|
| **Experimental models** | | |
| HEK-293T cells (*H. sapiens*) | Milani et al (2019) (Sci Transl Med) | N/A |
| Rat primary hippocampal neurons – CD (*R. Norvegicus*) | Charles River | N/A |
| C57BL/N Synapsin TKO (*M. musculus*) | Cambiaghi et al (2013) (Epilepsy Research) | N/A |
| C57BL/N (*M. musculus*) | Charles River | N/A |
| **Recombinant DNA** | | |
| MDLg/pRRE | Milani et al (2019) (Sci Transl Med) | N/A |
| pCMV.REV | Milani et al (2019) (Sci Transl Med) | N/A |
| pMD2.G | Milani et al (2019) (Sci Transl Med) | N/A |
| pAdVantage | Promega | E1711 |

| Reagent/Resource | Reference or source | Identifier or catalog number |
|---|---|---|
| pCCL-mCamKIIa(0.4)-EGFP (LV_EGFP) | This study | N/A |
| pCCL-mCamKIIa(0.4)-NPY-IRES-EGFP (LV_NPY) | This study | N/A |
| pCCL-mCamKIIa(0.4)-Y2R(flag)-T2A-NPY-IRES-EGFP (LV_Y2-NPY) | This study | N/A |
| **Antibodies** | | |
| βIII-Tubulin (mouse) | Covance | MMS-435P |
| Calnexin (rabbit) | Enzo life science | ADI-SPA-865-F |
| Flag-tag (mouse) | Sigma | 1804 |
| Glial fibrillary acidic protein (mouse) | Sigma | G3893 |
| Glutamate decarboxylase 1 (rabbit) | Cell signaling | 41318 |
| Glyceraldehyde-3-phosphate dehydrogenase (rabbit) | Sigma | G9545 |
| Goat-anti-chicken (488) | Invitrogen | A11039 |
| Goat-anti-chicken-IgG-HRP | Jackson Immuno Research | 103-035-155 |
| Goat-anti-mouse (594) | Invitrogen | A11005 |
| Goat-anti-mouse (647) | Invitrogen | A21235 |
| Goat-anti-mouse-IgG-HRP | Biorad | 1706516 |
| Goat-anti-rabbit (594) | Invitrogen | A11012 |

| Reagent/Resource | Reference or source | Identifier or catalog number |
|---|---|---|
| Goat-anti-rabbit (647) | Invitrogen | A21244 |
| Goat-anti-rabbit-IgG-HRP | Biorad | 1706515 |
| Green fluorescent protein (chicken) | Abcam | ab13970 |
| Neuropeptide Y (rabbit) | Cell signaling | 11976 |
| Parvalbumin (mouse) | Sigma | P3088 |
| Somatostatin (rat) | Synaptic System | 366 017 |
| Y2 Receptor (rabbit) | Neuromics | RA14112 |
| α-Tubulin (mouse) | Sigma | T9026 |
| **Oligonucleotides and other sequence-based reagents** | | |
| PCR Primers | This study | Table 1 |
| **Chemicals, enzymes and other reagents** | | |
| 30% acrylamide/Bis | Biorad | 1610158 |
| Agarose | Sigma | A9539 |
| AgeI-HF | NEB | R3552L |
| Amershan nitrocellulose | GE-Healthcare | 10600003 |
| Ammonium persulfate | Sigma | A3678 |
| B27 supplement | Thermo Fisher | 17504-044 |
| BCA Protein Assay Kit | Thermo Fisher | 23228 |
| Bovine Serum Albumin Fraction V | Roche | 10735086001 |
| $CaCl_2$ | Sigma | 202940 |
| D-glucose | Sigma | G5767 |
| DAKO Mounting Medium | Agilent | S3023 |
| DL-AP5 | Tocris | 0105 |
| DMEM high glucose | Sigma | D6546 |
| DNAse I (Bovine Pancreas) | Sigma | 260913 |
| DPBS | Euroclone | ECB4004L |
| EcoRV-HF | NEB | R3195S |
| Fetal Clone III | HyClone | SH30109.03 |
| Glutamate-Glo(TM) Assay, 5 ml | Promega | J7021 |
| Glycine | Sigma | G8898 |
| Hanks' Balance Salt Solution | Sigma | H9394 |
| HEPES | Sigma | H4034 |
| Iscove's Modified Dulbecco's Medium | Sigma | I3390 |
| KCl | Sigma | 409316 |
| $KH_2PO_4$ | Sigma | 7778-77-0 |
| Kynurenic acid | Sigma | k3375 |
| LB broth | Sigma | L3522 |
| LB-Agar | Sigma | L3147 |
| $MgCl_2$ | Sigma | 255777 |
| $MgSO_4$ | Sigma | 746452 |

| Reagent/Resource | Reference or source | Identifier or catalog number |
|---|---|---|
| MluI | Promega | R638A |
| NaCl | Sigma | 793566 |
| $NaH_2PO_4$ | Sigma | S5011 |
| $NaHCO_3$ | Sigma | 792519 |
| NheI-HF | NEB | R3131L |
| NucleoBond Xtra Maxi EF | Macherey-Nagel | 740424 |
| NuPAGE 12% bis-tris gel | Invitrogen | NP0341-box |
| Penicillin/Streptomycin | Sigma | P0781 |
| PFA 4% | ChemCruz | SC-281692 |
| Phusion-High Fidelity DNA Polymerase | NEB | M0350L |
| poly-L-ornithine | Sigma | P-3655 |
| Precision Plus protein Dual Xtra Standards | Biorad | 1610377 |
| Rat-Mouse NPY Elisa Kit | Sigma | EZRMNPY-27K |
| rSap | NEB | M0371S |
| SalI-HF | NEB | R3138S |
| SeeBlue Plus2 | Invitrogen | LC5925 |
| Sucrose | Sigma | S9378 |
| Supersignal West Femto Plus Chemiluminescence Substrate | Thermo Fisher | 34059 |
| Supersignal West Pico Plus Chemiluminescence Substrate | Thermo Fisher | 34580 |
| SYBR Safe DNA Gel Stain | Invitrogen | S33102 |
| T4 Ligase | NEB | M0202S |
| T4 PNK | NEB | M0201S |
| TEMED | Sigma | T7024 |
| Trizma-Base | Sigma | T6066 |
| Trypsin (Bovine Pancreas) | Sigma | T1005 |
| Trypsin-EDTA Solution | Sigma | 59417C |
| XbaI | NEB | R0145L |
| **Software** | | |
| SerialCloner 2-6-1 | http://serialbasics.free.fr/Serial_Cloner.html | |
| GraphPad Prism 10 | https://www.graphpad.com/ | |
| Fiji-ImageJ | https://imagej.net/software/fiji/ | |
| pClamp | https://www.moleculardevices.com/ | |
| ImageLab | https://www.bio-rad.com/ | |
| **Other** | | |
| Hamilton Syringe | Hamilton | 65460-05 |
| PhysioTel® ETA-F10 | DSI | 270-0160-001X |

## Materials

Cell culture media and reagents, if not otherwise stated, were from Thermo Fisher Scientific (Carlsbad, CA, USA). Plates and flasks were from Nalgene Nunc (Thermo Fisher Scientific). Petri dishes were from Falcon BD (Franklin Lakes, NJ, USA). Chemicals were from Tocris (Bristol, UK) or Merck-Sigma (Darmstadt, Germany).

## Antibodies

Details on the primary antibodies employed in Western Blot, Immunocytochemistry, and Immunohistochemistry analysis are reported in the Reagents and Tools Table. Secondary antibodies used for WB experiments were: Goat Anti-Mouse IgG (H+L)-HRP Conjugated (Biorad) 1:2000; Goat Anti-Rabbit IgG (H+L)-HRP Conjugated (Biorad) 1:2000; Peroxidase AfiiniPure Goat Anti-Chicken IgY (IgG) (H+L) (Jackson Immuno Research) 1:5000. Secondary antibodies used for immunochemistry experiments were: Alexa Fluor 647 Goat anti-mouse (Invitrogen; A21235); Alexa Fluor 594 Goat anti-rabbit (Invitrogen; A11012); Alexa Fluor 594 Goat anti-mouse (Invitrogen; A11005); Alexa Fluor 488 Goat anti-chicken (Invitrogen; A11039). Dilution of the secondary antibodies was: 1:250 for ICC and 1:300 for IHC.

## Primary culture of rat hippocampal neurons

The Institutional Animal Care and Use Committee of the San Raffaele Scientific Institute approved the animal use procedures. Primary cultures of hippocampal neurons were prepared according to (Bettegazzi et al, 2021b) from 2 to 3-day-old Sprague–Dawley rats. Briefly, after brain removal from the skull, and quick subdivision of hippocampi into small pieces, the tissue was incubated into Hank's solution containing 3.5 mg/mL trypsin type IX and 0.5 mg/mL DNase type IV for 5 min. The pieces were then mechanically dissociated in Hank's solution supplemented with 12 mM MgSO$_4$ and 0.5 mg/mL DNase IV. After centrifugation, cells were plated onto poly-ornithine coated coverslips and maintained in MEM supplemented with 20 mM glucose, B27, 2 mM glutamax, 5% serum, and 5 μM 1-β-D-cytosine-arabinofuranoside (Ara-C). Cultures were maintained at 37 °C in a 5% CO$_2$ humidified incubator.

## Plasmids

The pCCL-mCamKIIa(0.4)-EGFP plasmid was generated by cloning the mCamKIIa promoter sequence (kind gift of Dr. M. Ledri, Lund University). PCR-amplified sequence of the promoter was extracted with Primer #1 and #2 (Table 1) and subcloned into a lentiviral transfer plasmid pCCLsin.cPPT.PGK.GFP.wpre using EcoRV and AgeI.

To generate NPY-IRES-EGFP sequence fragment, adapter oligonucleotides (#3 and #4) were used to insert a NheI site through BamHI digestion ahead IRES sequence in a plasmid containing the IRES-EGFP sequence (the EMCV IRES from pIRES2-EGFP; Clontech). Adapters (#5 and #6) were used to insert a SalI site downstream EGFP coding sequence. The PCR-amplified sequence of murine NPY (Primer #7 and #8) has been extracted from a de novo synthesis plasmid containing both NPY and Y2R sequence (Gene Synthesis, Invitrogen) and subcloned

upstream to the IRES-EGFP through XbaI/NheI digestion. The pCCL-mCamKIIa(0.4)-NPY-IRES-EGFP was generated by cloning the NPY-IRES-EGFP sequence into the pCCL-mCamKIIa(0.4)-EGFP by substituting EGFP sequence through AgeI/SalI digestion.

To generate the Y2R(Flag)-T2A-IRES-EGFP sequence fragment, adapters (#9 and #10) were used to insert MluI and XbaI through AgeI/NheI digestion upstream of the IRES-EGFP sequence. Adapters (#11 and #12) were used to insert the entire sequence of the Flag-T2A fragment through MluI/XbaI digestion. Similar to NPY, the PCR-amplified sequence of the Y2R sequence (Primer #13 and #14) was extracted and subcloned upstream to the Flag-T2A-IRES-EGFP through AgeI/MluI digestion in frame with the Flag-tag. Subsequently, the Y2R(Flag)-T2A-IRES-EGFP fragment was cloned into the pCCL-mCamKIIa(0.4)-EGFP by AgeI/SalI digestion, removing the EGFP sequence. Finally, the pCCL-mCam-KIIa(0.4)-Y2R(flag)-T2A-NPY-IRES-EGFP was finally created by cloning the PCR-amplified sequence of NPY (Primer #7 and #8) downstream the T2A sequence in frame with Y2R by XbaI/NheI digestion.

All the steps requiring PCR amplification and restriction enzyme digestion were checked by Sanger Sequencing.

## LV production

Lentiviral vector production was performed as previously described (Milani et al, 2019). Briefly, HEK-293T cells (grown in IMDM medium (Merck-sigma) supplemented with 10% FBS, 2% L-glutamine, and 1% Pen/Strep) were transfected with packaging plasmids and the transfer vector plasmid of choice. After 30 h from transfection, cell culture supernatants were collected, filter-sterilized (0.22 μm), and centrifuged at 45,000×g for 2 h at 20 °C in a SW32.Ti rotor (Beckman Coulter). Pellets containing the viral particles were dissolved in PBS and stored at −80 °C until further use.

## LV titration

LV infectious titer was calculated by in vitro transduction of 293T cells and quantitative polymerase chain reaction (qPCR). Hundred thousand 293T cells were transduced with tenfold serial dilutions of LV prepared in fresh medium in the presence of polybrene (final concentration 8 μg/ml). Ten days after transduction, cells were collected to determine LV copies per diploid genome (vector copy number, VCN). For VCN determination, genomic DNA (gDNA) was extracted from transduced cells using Maxwell 16 DNA purification Kit (Promega) following the manufacturer's instructions. VCN was determined by analyzing 15–20 ng of gDNA through digital droplet PCR (ddPCR) with QX200 Droplet Digital PCR System (Biorad), according to the manufacturer's recommendations (each primer 900 nM, probe 250 nM). For quantification of LV genome copies, the following primers and a probe were used:

HIV Fw: 5′TACTGACGCTCTCGCACC-3′
HIV Rv: 5′- TCTCGACGCAGGACTCG-3′
HIV Pr 5′-(FAM)-ATCTCTCTCCTTCTAGCCTC-(MGBNFQ)-3′).

For quantification of human gDNA were used primers and probes designed on the GAPDH gene (Applied Biosystems Hs00483111_cn) or the human telomerase gene:

Telo fw: 5′-GGCACACGTGGCTTTTCG-3′;
Telo rv: 5′-GGTGAACCTCGTAAGTTTATGCAA-3′;

**Table 1. Primers designed for generating transfer plasmids used in the present work.**

| #N | Strand | Primer ID | Sequence |
|---|---|---|---|
| #1 | FW | EcoRV_mCamK | AAACGATATCTGACTTGTGGACTAAG |
| #2 | RW | EGFP | GAACTTGTGGCCGTTTAC |
| #3 | FW | NheI with BamHI | GATCCACGTGCTAGCACTGGC |
| #4 | RW | NheI with BamHI | GATCGCCAGTGCTAGCACGTG |
| #5 | FW | SalI with NotI | GGCCGTATTCAGGTCGACTAGGC |
| #6 | RW | SalI with NotI | GGCCGCCTAGTCGACCTGAATAC |
| #7 | FW | XbaI-AgeI-NPY | TTAATCTAGATACCGGTCCACCATGCTAGGTAACAAGC |
| #8 | RW | NheI-NPY | TAAGCTAGCTCATCACCAC |
| #9 | FW | AgeI-MluI-XbaI-NheI | CCGGTTACGCGACGCGTCGTAGGCTCTAGAGCTAAAG |
| #10 | RW | AgeI-MluI-XbaI-NheI | CTAGCTTTAGCTCTAGAGCCTACGACGCGTCGCGTAA |
| #11 | FW | (MluI) Flag-T2A (XbaI) | CGCGTGATTACAAGGATGACGACGATAAGGGCTCCGGCGAGGGCAGGGGAAGTCTTCTAACATGCGGGGACGTGGAGGAAAATCCCGGCCCAT |
| #12 | RW | (MluI) Flag-T2A (XbaI) | CTAGATGGGCCGGGATTTTCCTCCACGTCCCCGCATGTTAGAAGACTTCCCCTGCCCTCGCCGGAGCCCTTATCGTCGTCATCCTTGTAATCA |
| #13 | FW | AgeI-Y2R | GCAGCACCGGTCGCCACCATGGTTC |
| #14 | RW | MluI-Y2R | ATTTACGCGTCACATTGGTAGCC |

Telo probe: VIC 5′-TCAGGACGTCGAGTGGACACGGTG-3′ TAMRA.

To calculate VCN, we used the formula VCN = (ng LV)/(ng gDNA) × 2. An infectious titer is calculated with the formula transducing units (TU)/ml = (VCN × 100.000)/(1/(dilution factor)).

## LV transduction of primary rat hippocampal neurons

Primary rat neurons were transduced with a Multiplicity of Infection (MOI) of 3, with an adequate amount of viral vector particles (calculated considering number of cells to be transduced and vector titer), resuspended directly in culture medium. Primary hippocampal neurons were transduced at DIV4 and maintained at 37 °C, 5% $CO_2$ according to the experimental needs.

## Biochemical procedures

### Cell lysis

For NPY detection, primary neurons were lysed by direct addition of 2x sample buffer (100 mM Tris-HCl, pH 6.8, 5 mM EDTA/Na, 4% SDS, 10% glycerol, 0.4 M DTT, 0.02% bromophenol blue).

Mice hippocampi were homogenized in homogenization buffer (250 mM Sucrose, 2 mM EDTA/Na, 20 mM Hepes/Na pH 7.5, protease and phosphatase inhibitors) with 25 strokes of a glass-Teflon homogenizer and centrifuged at 500×g, 4 °C for 5 min. The supernatant S1 was then centrifuged at 100,000×g, 4 °C for 45 min. The supernatant was discarded, while the pellet, containing membranes, was resuspended in RIPA buffer (150 mM NaCl, 50 mM Tris-Cl (pH 8), 1% Tx-100, 0.5% Na-deoxycholate, and 0.1% SDS, protease and phosphatase inhibitors), incubated on ice for 15 min and then centrifuged at 10,000×g, 4 °C for 10 min. The protein content of the membrane extract was analyzed by BCA (Thermo Fisher Scientific, Waltham, MA, USA).

### Tissue homogenization

Mice hippocampi were homogenized in RIPA buffer (150 mM NaCl, 50 mM Tris-Cl (pH 8), 1% Tx-100, 0.5% Na-deoxycholate, and 0.1% SDS, protease and phosphatase inhibitors) with 25 strokes

of a glass-Teflon homogenizer and centrifuged at 15,000×g, 4 °C for 15 min, as in (Guarino et al, 2022). The protein content was analyzed by BCA (Thermo Fisher Scientific, Waltham, MA, USA).

### Western blot

As previously described (Bettegazzi et al, 2019), about 50 ug of proteins were separated by standard SDS-PAGE and transferred onto a nitrocellulose membrane. The nitrocellulose filter was stained with Ponceau S (0.2% in 3% trichloroacetic acid) and de-stained with double distilled water for protein visualization. After 1 h of blocking with TBST (10 mM Tris/HCl, 150 mM NaCl, 0.1% Tween-20) containing 5% bovine serum albumin (Roche Diagnostics, Basel, Switzerland) or skimmed powdered milk, the membranes were incubated overnight with the primary antibodies and, after extensive washing, with horseradish peroxidase-conjugated anti-rabbit or mouse secondary antibody (Bio-Rad, Hercules, CA, USA). For loading controls, membranes were stripped in an acidic buffer (0.2 M glycine, 0.1% SDS, 1% Tween-20, pH 2.2) and re-probed with the appropriate antibody. Proteins were revealed by direct acquisition using the Biorad Chemidoc Imaging system by Super Signal West Chemiluminescent Substrate (Thermo Fisher Scientific). Bands were quantified using ImageJ, and protein levels were normalized against the loading control.

### Immunocytochemistry on a glass slide

Neurons were grown on glass coverslips in vitro, fixed with 4% paraformaldehyde solution, containing 4% sucrose in PBS. Primary antibodies were incubated 1h at RT, diluted in blocking solution (1% normal goat serum/0.1% Triton in PBS), followed by 1h incubation with fluorescent secondary antibodies, as in (Bettegazzi et al, 2021a). Afterward, coverslips were incubated with DAPI for 5' at RT, deepen in water and mounted on microscope slides with Fluorescence Mounting Medium (DAKO, Agilent).

### ELISA assay

Total secreted NPY, in the culture media of primary rat hippocampal neurons, was evaluated by the rat specific ELISA assay (Merck-Sigma) according to the manufacturer's instructions.

### Glutamate release

Neurons used for glutamate measurement were transduced with the indicated vectors at DIV4 (MOI5), and experiments were performed at 13–15 DIV. Culture medium was substituted with Krebs Ringer Hepes buffer without magnesium (KRH w/o Mg, containing 5 mM KCl, 125 mM NaCl, 2 mM $CaCl_2$, 1.2 mM $KH_2PO_4$, 6 mM glucose, and 20 mM Hepes, pH 7.4) and cells were incubated for 20 min at 37 °C before collection of supernatant and lysis. Neuronal glutamate release was measured using the Glutamate-Glo Assay (Promega, Madison, Wisconsin, USA), according to the manufacturer's instructions. Briefly, an equal amount of sample and glutamate detection reagent (containing glutamate dehydrogenase, NAD+, reductase, reductase substrate, and luciferase) were incubated at room temperature for 2 h before reading the luminescent signal with a Victor 3 Luminometer (Perkin Elmer, Waltham, Massachusetts, USA). The luminescent signal, that is proportional to the amount of glutamate in the sample, was normalized to the total protein content of each sample and represented as fold change over control values.

### Electrophisiology

Primary culture slides of rat hippocampal neurons transduced with LV vectors were submerged in a recording chamber mounted on the stage of an upright BX51WI microscope (Olympus, Japan) equipped with differential interference contrast optics (DIC). Slides were continuously perfused with artificial cerebrospinal fluid (ACSF) containing (in mM): 125 NaCl, 2.5 KCl, 1.25 $NaH_2PO_4$, 2 $CaCl_2$, 25 $NaHCO_3$, 1 $MgCl_2$, and 11 D-glucose saturated with 95% $O_2$, 5% $CO_2$ (pH 7.3) flowing at a rate of 2–3 ml/min at room temperature. Whole-cell patch-clamp recordings were performed using glass pipettes filled with a solution containing (in mM): 30 $KH_2PO_4$, 100 KCl, 2 $MgCl_2$, 10 NaCl, 10 HEPES, 0.5 EGTA, 2 $Na_2$-ATP, 0.02 Na-GTP (pH 7.2, adjusted with KOH; tip resistance: 6–8 MΩ).

All recordings were performed using a MultiClamp 700B amplifier interfaced with a PC through a Digidata 1440A (Molecular Devices, Sunnyvale, CA, USA). Data were acquired and analyzed using the pClamp10 software (Molecular Devices). Voltage- and current-clamp traces were sampled at a frequency of 30 kHz and low-pass filtered at 2 kHz.

## Animals

Experiments were carried out in male TKO mice for *Syn1*, *Syn2*, and *Syn3* (Boido et al, 2010) and in WT mice with C57BKL/6 N genetic background. Mice were housed under controlled temperature (22 ± 1 °C) and humidity (50%) conditions following a 12 h light/dark cycle. Food and water were provided ad libitum. Mice were maintained and bred at the animal house of Ospedale San Raffaele in compliance with institutional guidelines and international laws (EU Directive 2010/63/EU EEC Council Directive 86/609, OJL 358, 1, December 12, 1987, NIH Guide for the Care and Use of Laboratory Animals, US National Research Council, 1996; authorization n. 1090/2020-PR, and 1079 INTEG. ISS 3). All efforts were made to minimize animal suffering.

### Intracardial perfusion, brain collection, and fixation

Mice intended to immunohistochemistry examination were anesthetized with an intraperitoneal injection of a mixture of ketamine/xylazine (100 and 10 mg/kg, respectively). Transcardial perfusion was performed with 25 ml of ice-cold 1x phosphate-buffered saline (PBS) and subsequent 25 ml of 4% paraformaldehyde in PBS (pH 7.4). After decapitation, the brain was rapidly removed from the skull and kept in 4% paraformaldehyde in PBS for 16 h at 4 °C. Brains were then washed three times with 1x PBS for 10 min and cryoprotected in 30% sucrose in 1x PBS for 2 days. Brains were rapidly frozen and 15-μm-thick coronal sections were cut with a CM3050s cryostat (Leica Microsystems) and placed onto SuperFrost slides, allowed to dry at room temperature, and kept at −20 °C for further procedures.

### Immunofluorescence on brain slices

Slices were incubated with a blocking solution containing 1% normal goat serum (NGS), and 0.3% Triton in 1x PBS for 1 h at 4 °C. Subsequently incubation with primary antibodies diluted in the same solution were carried overnight at 4 °C. Slices were rinsed two times with 15 min of 1x PBS and 15 min of blocking solution kept at 4 °C. Incubation with secondary antibodies diluted in the blocking solution was performed at room temperature for 1.5 h in a dark room. Appropriate fluorophore-conjugated (Alexa Fluor® 488, 594, 647; Molecular probes) secondary antibodies were used according to the manufacturer's instructions. Slices were then rinsed three times with 15 min of 1x PBS and 15 min of blocking solution, incubated for 5 min with 5 mg/ml DAPI (Merck-Sigma), and mounted with DAKO mounting medium (Agilent).

Immunofluorescence with Y2R antibody required heat-induced antigen retrieval. Thus, prior to the described immunofluorescence protocol, brain slices positioned onto super frost glass slides were placed in a tray containing sodium citrate buffer (10 mM Sodium Citrate; pH 6.0). The buffer tray was heated in a microwave, heating was stopped before reaching boiling temperature, and the buffer was let chill at RT for 30' before proceeding.

### Imaging and analysis

All the confocal images have been acquired in the Advanced Light and Electron Microscopy BioImaging Center (Alembic). We followed a strictly pre-determined procedure. Images were acquired in an 8-bit format at a resolution of 1024 × 1024 pixels. For all images, we used a Leica SP8 confocal system equipped with an acousto-optical beam splitter which allows the selection of a specific frequency range for every fluorophore. Laser intensity and offset were differentially determined for every antibody and then maintained identically across samples to allow a qualitative comparison. The one and only exception was made for the laser intensity used to detect the EGFP signal due to the presence of an IRES sequence. Thus, laser intensity was increased in the samples treated with either mCamK-NPY-IRES-EGFP and mCamK-Y2-NPY-IRES-EGFP compared to controls. Analysis and acquisition of fluorescence images could not be done blindly, because the EGFP signal was obviously lower in the groups treated either with mCamK-NPY-IRES-EGFP or mCamK-Y2-NPY-IRES-EGFP. However, the EGFP signal in this setting is only used to visualize targeted cells, and no quantifications were envisaged.

Image analysis was performed with Fiji-ImageJ software. Analysis of the co-localization of the inhibitory neuronal markers in the LV_EGFP transduced mice was done by visually identifying

positive cells and manually quantifying double positive interneurons. For binary processed images only: after the subtraction of the same arbitrary threshold, images were transformed into binary to visualize the difference more clearly.

### Image composition

Image compositions were made with Adobe Illustrator cc 2017 (Adobe System, San Jose, CA, USA). Drawings were created with BioRender.com under a license granted to M.S.

## Surgical procedures

All the surgical procedures were performed on anesthetized mice placed in a stereotaxic frame (Stoelting Co.). Anesthesia was induced in a plastic chamber saturated with a mixture of 3% isoflurane/0.5% $O_2$ (Harvard apparatus). Subsequently, mice were positioned in the stereotaxic frame with the nose inserted into a mask for anesthesia where a mixture of 2% isoflurane/1% $O_2$ was continuously flowing for the entire time of the procedure.

### Viral vector stereotaxic injection in the mouse hippocampus

Mice used for biochemical and histological examination were injected with viral vectors at 60 days of age. TKO littermates were randomly distributed in the two groups. TKO mice used for the video-EEG experiment were injected with viral vectors at 45 days of age. In both conditions, after quick shaving of the head of the mouse, the skull of mice steadily positioned in the stereotaxic frame was exposed through a cut along the anteroposterior axis. A drill was used to prepare a burr hole at coordinates: AP: −1.8; ML: ± 1.8; DV: −2.0. The coordinates were measured in relation to bregma after the alignment of the bregma-lambda axis. Mice intended for histological examination were injected homolaterally. Mice intended for biochemical analysis and animals that underwent video-EEG monitoring were injected bilaterally.

The infusion was performed through a 34G stainless steel needle (Hamilton company) connected to a 25 μL Hamilton syringe. The syringe was mounted onto a peristaltic pump (Legato® 130 syringe pump; KD scientific) positioned directly on the arm of the stereotaxic frame. Two μl of viral vector preparation were infused at a rate of 200 nl/min (10 min total). After the injection, the needle was left in place for 5 additional minutes to allow the diffusion of vector particles before being slowly withdrawn from the brain. Off-target injection of the vector was considered as an exclusion criterion. However, this event never occurred in the present experimental cohort.

### Telemetry transmitter implant for video-EEG monitoring

TKO mice intended for video-EEG experiment underwent surgery for transmitter implant between 52 and 55 days of age. The ETA-F10 transmitter (Data Sciences International, St. Paul, MN) was positioned in a subcutaneous pocket on the back of the mouse with the wires guided to the skull. The recording electrode was placed on the *dura mater* above the hippocampus, and the reference electrodes were placed contralaterally on the *dura mater*, anterior to the bregma. Once the electrodes were in position, dental cement (Harvard Apparatus) was added to cover and attach the implant to the skull. Recordings started when animals were at 60 days of age and lasted for 19 days. After LV injection and EEG-telemetry probe implantation, animals were randomly distributed in the monitoring cages, and the analysis of seizure occurrence was performed blindly by the operator.

### Video-EEG monitoring

Neuroscore (Data Sciences International, St. Paul, MN) was used for EEG analysis. All traces were visually inspected for the detection of seizures and duration was measured as the period of paroxysmal activity of high frequency (>5 Hz) characterized by a threefold higher amplitude over baseline with a minimum duration of at least 5 s.

## Statistical analysis

Statistical analysis was performed with GraphPad Prism software version 9.0 (GraphPad Software Inc., La Jolla, CA, USA). Results are given as dot plots or histograms with mean ± SEM as specified in the figure legends. A number of experiments and *p* value are indicated in the figure legends, while biological replicates are shown as individual points in graphs. To assess for normal distribution of the data we applied the normality test (D'Agostino-Pearson) and equal variance test (Brown-Forsythe). In the case of normally distributed data, statistical significance was evaluated (with 95% confidence intervals) by one-way ANOVA followed by Dunnett post hoc test (for multiple comparisons against a single reference group), Newman–Keuls or Tukey's post hoc test (for multiple comparisons between groups), two-tailed Student *t*-test (for comparisons between two average values); for samples with non-normal distributions, the nonparametric Mann–Whitney *U*-test (for significant differences between two experimental groups) and the Kruskal–Wallis one-way analysis of variance followed by Dunn's post hoc test (for the analysis of multiple experimental groups) were used. A value of $P < 0.05$ was considered statistically significant. The sample size was calculated using G Power software (version 3.1), based on effect sizes calculated from our preliminary experiments, with a power of 0.8 and alpha = 0.05.

## Data availability

This study includes no data deposited in external repositories.

The source data of this paper are collected in the following database record: biostudies:S-SCDT-10_1038-S44319-024-00244-0.

## Peer review information

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

## Acknowledgements

The authors thank the ALEMBIC facility at the San Raffaele Scientific Institute for imaging. We also thank Elena Monzani for the technical assistance with mouse breeding. This work was supported by a grant from the European Community (FP7- HEALTH Project 602102 [EPITARGET]).

## Author contributions

**Stefano Cattaneo**: Conceptualization; Data curation; Formal analysis; Investigation; Methodology; Writing—original draft; Writing—review and editing. **Barbara Bettegazzi**: Conceptualization; Data curation; Formal analysis; Investigation; Methodology; Writing—original draft; Writing—review and editing. **Lucia Crippa**: Investigation; Methodology. **Laila Asth**: Investigation; Methodology. **Maria Regoni**: Investigation; Methodology. **Marie Soukupova**: Investigation; Methodology. **Silvia Zucchini**: Supervision; Investigation; Project administration. **Alessio Cantore**: Supervision; Methodology. **Franca Codazzi**: Investigation; Methodology. **Flavia Valtorta**: Supervision; Investigation. **Michele Simonato**: Conceptualization; Supervision; Funding acquisition; Writing—original draft; Project administration; Writing—review and editing.

Source data underlying figure panels in this paper may have individual authorship assigned. Where available, figure panel/source data authorship is listed in the following database record: biostudies:S-SCDT-10_1038-S44319-024-00244-0.

## Disclosure and competing interests statement

The authors declare no competing interests.

# Expanded View Figures

**Figure EV1. Electrophysiological characterization of LV transduced primary rat hippocampal neurons.**

(A–C) Voltage response of LV transduced neurons to suprathreshold depolarizing currents steps. (D–H) Dot plots summarizing average resting membrane potential, action potential threshold, input resistance, spontaneous excitatory post-synaptic currents (sEPSCs) amplitude, and sEPSCs frequency of transduced neurons. Data were shown as mean ± SEM of 3 independent experiments (biological replicates). (D) LV_EGFP = −55.46 ± 1.77 mV; LV_NPY = −57.08 ± 2.07 mV; LV_Y2-NPY = −56.43 ± 2.07 mV. (E) LV_EGFP = −32.64 ± 0.45 mV; LV_NPY = −33.00 ± 0.66 mV; LV_Y2-NPY = −32.21 ± 1.4 mV. (F) LV_EGFP = 261.4 ± 20.94 MΩ; LV_NPY = 223.8 ± 44.47 MΩ; LV_Y2-NPY = 271.5 ± 42.14 MΩ. (G) LV_EGFP = 30.93 ± 4.12 pA; LV_NPY = 28.17 ± 3.25 pA; LV_Y2-NPY = 34.17 ± 8.49 pA. H, LV_EGFP = 2.12 ± 0.38 Hz; LV_NPY = 1.69 ± 0,32 Hz; LV_Y2-NPY = 2.63 ± 0.66 Hz. Source data are available online for this figure.

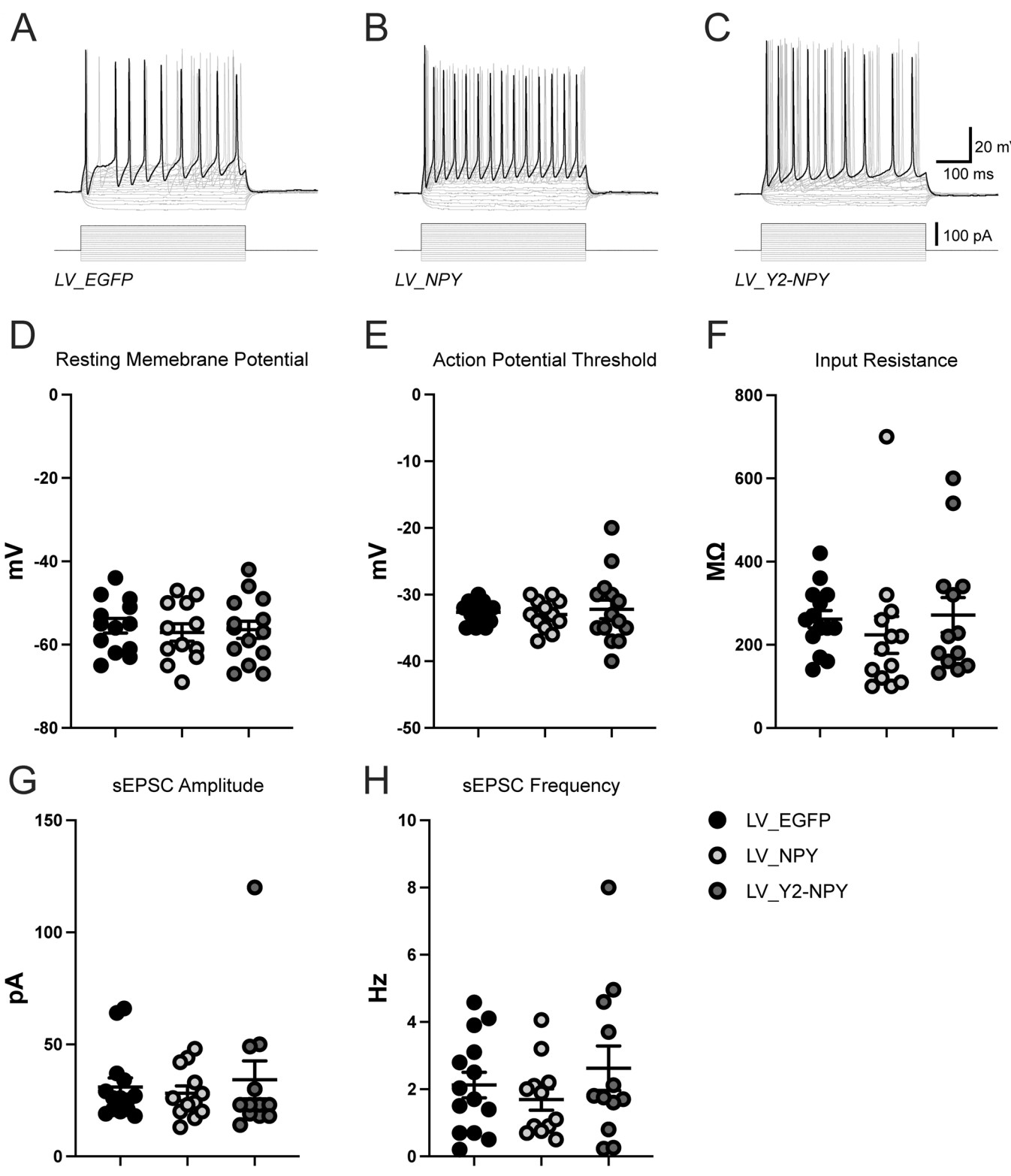

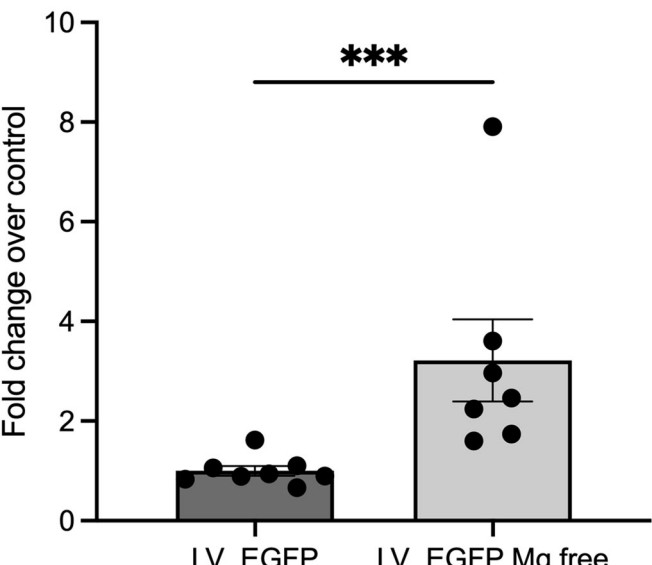

**Figure EV2.   Effect of Mg²⁺ removal on glutamate release.**

Measurement of extracellular glutamate in transduced primary hippocampal neurons incubated for 20 min in Kreb's Ringer Hepes in the presence (LV_EGFP) or absence of Mg²⁺ (LV_EGFP Mg free). Values were normalized for total protein content and are presented as means ± SEM of at least three independent experiments (biological replicates), with glutamate levels shown as fold change over control (LV_EGFP = 1.00 ± 0.10; LV_ EGFP Mg free = 3.22 ± 0.82; $p$ = 0.0006). Statistical significance was calculated using the Mann–Whitney test. ***$p$ < 0.0001. Source data are available online for this figure.

