## [Peer Review File · EMBO Reports]

Gene therapy for epilepsy targeting Neuropeptide Y and its Y2 receptor to dentate gyrus granule cells

Stefano Cattaneo, Barbara Bettegazzi, Lucia Crippa, Laila Asth, Maria Regoni, Marie Soukupova, Silvia Zucchini, Alessio Cantore, Franca Codazzi, Flavia Valtorta, and Michele Simonato

Corresponding author(s): Michele Simonato (simonato.michele@hsr.it)

Review Timeline:

Submission Date:	20th Dec 23
Editorial Decision:	26th Jan 24
Revision Received:	12th Jul 24
Editorial Decision:	2nd Aug 24
Revision Received:	15th Aug 24
Accepted:	22nd Aug 24

Editor: Esther Schnapp

Transaction Report: This manuscript was transferred to EMBO reports following peer review at EMBO Molecular Medicine.

Referee #1

There are several papers showing an anticonvulsive effect of combined injections of NPY and Y2 vectors in other animal models. This paper, however uses CamKinase IIa promoters to allow expression only on glutamatergic neurons.

The authors use intrahippocampal injections of NPY and Y2 receptor vectors on CamKinase IIa promoter and test this in on epileptic activity in synapsin triple KO mice.

This approach has been well established (several times) for NPY vectors and NPY + Y2 vectors in other animal models and using other promoters. The use of CamKinase II promoters used here, however, has the purpose to restrict expression to glutamatergic neurons.

Although the findings are not entirely novel, the paper shows a very nice analysis of the sites of vector expression and documents the anti-seizure effect of a combination of Y2 and NPY expressing vectors.

The experiments document expression of the vectors mainly (or exclusively) in mossy fibers. Surprisingly, they are obviously not expressed in mossy cells of the dentate gyrus. Why? Otherwise one would find NPY and Y2 in the inner molecular layer.

Figure 5. What does the difference in NPY expression between B and D mean? At p100 there was a high expression of NPY in mossy fibers (B), not at p120? How does NPY expression in mossy fibers relate to recent seizure activity? Is this expression typical for all animals?

Figure 6. To which time interval does "number of seizures" refer to? Video recording started two weeks after vector injection. How long did the recording last?

6E. Would in the Y2-NPY group the number of seizure-free mice go further down after 30 days? I think one should investigate this.

Referee #2

The choice of Synapsin TKO as model raises questions. It is not a recognized model of drug-resistant epilepsy, and its translational potential is unclear. The authors should have used other models where AAV-NPY-Y2 has already been tested

Cattaneo and colleagues devised a combinatorial approach involving lentivirus delivery of NPY and its receptor Y2, driven by a CamkII promoter. They demonstrated NPY and Y2 expression both in vitro and in vivo. Additionally, they observed no expression in astrocytes and inhibitory neurons in vivo. Subsequently, they administered this lentiviral tool to Synapsin TKO mice

before the onset of seizures, revealing a reduction in seizures and an increase in seizure-free animals.

To the best of my knowledge, this marks the first gene therapy approach for Synapsin TKO mice. Nevertheless, caution is necessary before characterizing it as a "promising treatment option for drug-resistant epilepsy." Several major concerns arise:

1. The statement, "However, none so far included a built-in auto-regulatory system, and each, therefore, may lead to excessive inhibition," is inaccurate. Previous auto-regulatory gene therapies for epilepsy, such as those conducted by Kullmann's lab at UCL, and those including work on NPY and Y2 co-expression using a single AAV (e.g., Melin et al., 2019, 2023, and Szczygie et al., 2020).

2. The choice of Synapsin TKO as model raises questions. It is not a recognized model of drug-resistant epilepsy, and its translational potential is unclear. Why did the authors not opt for other models where AAV-NPY-Y2 has already been tested?

3. In terms of translation, lentiviral delivery in the human hippocampus may not be a realistic therapy due to the limited virus spread. Why was AAV not used, especially considering its previous use with NPY-Y2? The novelty here seems to lie in the specific excitatory promoter.

4. The specificity of CamKII to excitatory neurons, particularly the lack of proper quantification, including inhibitory neurons and the percentage of transduced cells, needs clarification. Literature reports mixed results.

5. Could the observed effect be a mere delay in epileptogenesis? A longitudinal study is crucial to explore this possibility, as the two-week recording period appears short given the dynamics shown in Figure 6E.

6. The use of T2A for NPY-Y2, as previously done by Melin et al., renders the discussion on this matter redundant. It would be more beneficial to focus on how this therapy fits into the context of other developed gene therapies.

7. Although the authors discussed the self-regulated approach, direct evidence supporting this claim is lacking in the present work.

8. If the authors aim to establish the superiority of this therapy over NPY alone, the addition of another group to the seizure analysis for a direct comparison is essential

Referee #3

The manuscript describes the result of a newly developed lentiviral vector which enables co-expression of anti-convulsant neuropeptide NPY and its autoreceptor Y2 genes in the epileptic hippocampus, and evaluates its ability to suppress the seizures *in vivo*. The critical feature of this viral vector is its use of CamKIIa promoter driving both transgenes in the same excitatory neurons, which allows autoregulation of neural excitability through Y2 receptor-mediated inhibition. Telemetry video-EEG monitoring was conducted to detect seizures in the TKO mice expressing NPY/Y2 transgenes in dentate gyrus granule cells, and a clear reduction of the seizure occurrence in these mice was demonstrated.

This is a well thought out and seems to be a well carried out study. The idea of "autoregulation" of the epileptic brain activity is novel and very interesting. The choice of the model organism

(i.e., mouse) is adequate, the statistical analysis methods used are adequate, and their usage of lentiviral vector for gene therapy seems to be reasonable and less hazardous compared to other viral vectors.

The main concern I have is that a few important control experiments are missing in this study. It weakens the overall impact of the core message "autoregulation of the epileptic activity". I would like the authors discuss more thoroughly the mechanism how the autoregulation is realized in their model system to strengthen the core message.

Major points;

1) "Necessity of combinatorial gene expression"

Authors demonstrated that expressing NPY and Y2R transgenes suppressed the seizures very well (Fig.6) and argue that the combinatorial expression is important. Unfortunately, however, the authors didn't show that expressing either of the two genes alone is NOT sufficient for suppressing seizures. Induction of the Y2R gene alone in the excitatory neurons may be sufficient for suppressing seizures, considering the low but constitutive expression of the endogenous NPY peptides in the hippocampus, and the increase in its expression level at the period of seizure onset (Fig. 5A). If that is the case, expressing a single gene, for example, a constitutively active form of the Y2R gene alone in place of the wild-type counterpart, seems to be a simpler and more secured way to suppress the seizures.

Forced expression of NPY gene in the excitatory neurons may also be sufficient. At least, authors could have examined the effect from NPY alone (LV_NPY) in vivo and evaluated its efficacy of suppression. I would like the authors to add a few control experiments to strengthen the argument that the combinatorial gene expression is critical.

2) "Autoregulation on the seizure activity"

Authors argue that combinatorial expression of the NPY and Y2R genes in Hippocampal excitatory neurons exerts an "auto-feedback" regulation on the epileptic activity. When it comes to autoregulation, one would imagine a series of consecutive events in the hippocampus; 1) NPY release triggered by a spontaneous epileptic activity in the excitatory neurons, 2) Y2R activation by the induced NPY, 3) hyperpolarization of the membrane potential due to the K-channel opening after Y2R activation or whatever, and 4) premature shut-down of the epileptic activity. If the scenario is right, frequency of the seizure events in TKO mice would not be influenced much by expressing the NPY-Y2R transgenes, but the seizure length should be shortened drastically. The results shown in figure 6 C,D, however, are not consistent with the scenario. Detectable seizure was nearly absent in TKO/LV_NPY_Y2R mice. I wonder how and when the NPY-Y2R system was activated without triggered by seizure activity in these seizure-free animals. I'd like the authors discuss more about the mechanism how the NPY-Y2R system autoregulates the spontaneous seizures in TKO mice.

Another scenario is that TKO/LV_NPY_Y2R mice still had sub-threshold (undetectable) seizure activities frequently, which continuously activated the NPY/Y2R system in these mice. If that is the case, expressing a constitutively active form of the Y2R gene alone would be sufficient for suppressing the seizures.

3) "interference on the normal hippocampal function"

I agree with the authors that forced expression of the NPY-Y2R transgenes in the hippocampal

excitatory neurons suppress the seizures quite well (Fig. 6). Now, I would like the authors to discuss the safety of the LV_NPY_Y2R.

It would be very nice if the authors can show that forced expression of the NPY-Y2R genes does not perturb the normal activity of the healthy excitatory neurons. Did authors observe any sign of behavioral abnormality in LV_NPY_Y2R mice? Authors should report detailed non-ictal behaviors/observations of the TKO/LV_NPY_Y2R mice. Autoregulating the epileptic activity as necessary seems to be definitely a better strategy than continuously lowering the neural activity using a constitutively active form of Y2R gene, which presumably has deleterious effects on the normal function of the hippocampus. Authors should discuss the merits and demerits of combinatorial gene expression strategy in comparison with single gene strategy, such as a constitutively active form of Y2R.

Minor points;

Fig.2 D. The bands for pre-processed NPY and mature NPY should be annotated and indicated by arrows. The unit (kDa ?) should also be indicated. The two lanes for LV_Y2-NPY are redundant.

Fig6 E,F. Did the authors also check whether the NPY/Y2R transgenes were successfully expressed or not in the TKO/LV_NPY_Y2R mice with seizures (n = 4)? Would the failure of suppressing the seizure in these mice be accounted for simply by a failure of NPY/Y2R transgene expression in the hippocampus?

The success of the combinatorial gene therapy using NPY-Y2R system is clear. The manuscript as in the current form is worth while publishing at least as a short report. For a full report, authors may want to consider the concerns above, and revise the manuscript.

Cattaneo et al.

“Combinatorial gene therapy for epilepsy based on Neuropeptide Y and its Y2 receptor”
Ms number **EMBOR-2023-58691-T** (transfer from EMBO Mol Med EMM-2023-18970)

Point-by-point response to Reviewers

Referee #1 (Comments on Novelty/Model System for Author):

There are several papers showing an anticonvulsive effect of combined injections of NPY and Y2 vectors in other animal models. This paper, however uses CamKinase IIa promoters to allow expression only on glutamatergic neurons.

Referee #1 (Remarks for Author):

The authors use intrahippocampal injections of NPY and Y2 receptor vectors on CamKinase IIa promoter and test this in on epileptic activity in synapsin triple KO mice.

This approach has been well established (several times) for NPY vectors and NPY + Y2 vectors in other animal models and using other promoters. The use of CamKinase IIa promoters used here, however, has the purpose to restrict expression to glutamatergic neurons.

Although the findings are not entirely novel, the paper shows a very nice analysis of the sites of vector expression and documents the anti-seizure effect of a combination of Y2 and NPY expressing vectors.

The experiments document expression of the vectors mainly (or exclusively) in mossy fibers. Surprisingly, they are obviously not expressed in mossy cells of the dentate gyrus. Why? Otherwise one would find NPY and Y2 in the inner molecular layer.

RE: Indeed, some degree of expression in mossy cells was detected, with labeling of the inner molecular layer, which was more obviously detectable contralaterally. This information was reported and shown in the original manuscript (page 6, first 2 lines, and Fig. 3D).

ACTION: To refine this observation, we may perform immunofluorescence labeling of mossy cells in LV_EGFP animals using an antibody against mGluR2/3 (Bui et al., Science 2018).

Figure 5. What does the difference in NPY expression between B and D mean?

RE: Panel B represents the quantitative measurements of NPY hippocampal levels that were performed in 5 animals per group per time point at four different timepoints (p30/60/90/120). Panel D is a representative image that qualitatively indicates the localization of NPY upregulation, that is, mainly in the mossy fiber terminals of the TKO mice.

At p100 there was a high expression of NPY in mossy fibers (B), not at p120?

RE: Indeed, we detected increased NPY expression at p90, when most of the animals were experiencing spontaneous seizures, while NPY levels returned to baseline by p120. In other words, increased NPY expression follows the onset of spontaneous seizures and precedes their attenuation. This suggests that NPY overexpression may represent an adaptive mechanism to seizure activity, that is set in place to counteract seizures themselves (which in fact decrease after p90 in TKO mice).

How does NPY expression in mossy fibers relate to recent seizure activity?

RE: In keeping with findings in other models (Vezzani et al., Neuropeptides 2004; Cattaneo et al., Front Mol Neurosci 2021), we hypothesize that reactive NPY overexpression reflects

intensity of seizure activity also in the TKO model. However, we did not measure the correlation between seizure activity and NPY expression, as this was beyond the goals of the present study.

ACTION: This point may be briefly discussed in the revised manuscript.

Is this expression typical for all animals?

RE: Yes, this pattern is observed in all animals that were analyzed.

Figure 6. To which time interval does "number of seizures" refer to? Video recording started two weeks after vector injection. How long did the recording last?

RE: The time interval of "number of seizures" refers to the three weeks of recording.

ACTION: This may be better specified in the figure legend.

6E. Would in the Y2-NPY group the number of seizure-free mice go further down after 30 days? I think one should investigate this.

RE: We agree that it would be interesting to know whether seizure freedom is maintained for a longer time. Unfortunately, we could not prolong the recording period over 3 weeks because of limited duration of the telemetry battery. Please remember that the cost of recording animals in the telemetry system is very high. We are aware of the need of refining our observations in the future, in longer time periods and in other epilepsy models.

ACTION: We will explicitly state this need in the revised manuscript.

Referee #2 (Comments on Novelty/Model System for Author):

The choice of Synapsin TKO as model raises questions. It is not a recognized model of drug-resistant epilepsy, and its translational potential is unclear. The authors should have used other models where AAV-NPY-Y2 has already been tested

RE: The multiple reasons why we selected the TKO model for this study are described in the original manuscript (page 6, last 3 lines to page 7, end of first paragraph). As stated above, while we are aware of the need of extending our observations to other models, we thought that this was a good model to start testing our new vector. Incidentally, please note that this point was raised only by Referee #2, whereas Referee #1 did not comment on it, and Referee #3 explicitly stated that "The choice of the model organism is adequate".

Referee #2 (Remarks for Author):

Cattaneo and colleagues devised a combinatorial approach involving lentivirus delivery of NPY and its receptor Y2, driven by a CamkII promoter. They demonstrated NPY and Y2 expression both in vitro and in vivo. Additionally, they observed no expression in astrocytes and inhibitory neurons in vivo.

Subsequently, they administered this lentiviral tool to Synapsin TKO mice before the onset of seizures, revealing a reduction in seizures and an increase in seizure-free animals.

To the best of my knowledge, this marks the first gene therapy approach for Synapsin TKO mice. Nevertheless, caution is necessary before characterizing it as a "promising treatment option for drug-resistant epilepsy."

RE: We agree with the Referee. Our statement in the Abstract was excessively optimistic, while our view is better expressed elsewhere in the manuscript.

ACTION: We will rephrase this sentence appropriately.

Several major concerns arise:

1. The statement, "However, none so far included a built-in auto-regulatory system, and each, therefore, may lead to excessive inhibition," is inaccurate. Previous auto-regulatory gene therapies for epilepsy, such as those conducted by Kullmann's lab at UCL, and those including work on NPY and Y2 co-expression using a single AAV (e.g., Melin et al., 2019, 2023, and Szczygie et al., 2020).

RE: We agree that our statement is inaccurate. Please note, however, that whereas the Kullmann's lab (Qiu et al., Science 2022) proposed a regulated system based on the expression of therapeutic genes under the control of an activity-dependent promoter (that is, a regulation at transcriptional level), our approach entails an autoregulation in which transgenes are continuously expressed, not altering normal neuronal function but activating an auto-regulatory loop in case of hyper-activity. With reference instead to previous studies on NPY/Y2, as recognized by all Referees, the novelty is the expression of transgenes in excitatory neurons by using the CaMKII promoter (by the way, this is a necessary prerequisite for our auto-regulatory strategy). In sum, we believe that it is fair to say (words of Reviewer #3) that the study "can be another excellent work supporting the concept in a complementary fashion".

ACTION: We will revise the manuscript in order to make the novelty clearer, with reference in particular to the studies mentioned by the Referee.

2. The choice of Synapsin TKO as model raises questions. It is not a recognized model of drug-resistant epilepsy, and its translational potential is unclear. Why did the authors not opt for other models where AAV-NPY-Y2 has already been tested?

Please see above, our response to the Comments on Novelty of this Referee.

RE: The multiple reasons why we selected the TKO model for this study are described in the original manuscript (page 6, last 3 lines to page 7, end of first paragraph). As stated above, while we are aware of the need of extending our observations to other models, we thought that this was a good model to start testing our new vector.

Incidentally, please note that this point was raised only by Referee #2, whereas Referee #1 did not comment on it, and Referee #3 explicitly stated that "The choice of the model organism is adequate".

ACTION: We will explicitly state in the revised manuscript that it will be needed in the future to extend our observations to other models.

3. In terms of translation, lentiviral delivery in the human hippocampus may not be a realistic therapy due to the limited virus spread. Why was AAV not used, especially considering its previous use with NPY-Y2? The novelty here seems to lie in the specific excitatory promoter.

RE: The choice of primarily using lenti- rather than AAV vectors was indeed based on the idea to use a specific excitatory promoter (CamKII), which implies a relatively large expression cassette that could hardly fit in AAV. In addition, we aimed at restricting the expression, for

what possible, to granule cells. While AAV vectors certainly ensure a greater spread, one should also take into account that they must be used at high titers, which raise some concerns (e.g., BBB-breakdown and DNA interference - Guo et al., Mol Ther Meth & Clin Devel 2023). A comment of Referee #3 is perfectly in line with these considerations: “usage of lentiviral vector for gene therapy seems to be reasonable and less hazardous compared to other viral vectors”. Note also that a currently approved gene therapy clinical trial for focal epilepsy uses lentivectors (ClinicalTrials.gov ID NCT04601974); thus, it is not demonstrated that the limited spread of the virus cannot ensure a therapeutic effect.

ACTION: We may better explain the reasons for choosing a lentivector platform for the present study.

4. The specificity of CamKII to excitatory neurons, particularly the lack of proper quantification, including inhibitory neurons and the percentage of transduced cells, needs clarification. Literature reports mixed results.

RE: We have tested the specificity of CamKII in excitatory neurons by verifying lack of expression in either parvalbumin-positive and NPY-positive inhibitory neurons, and in astrocytes (Fig. 3).

ACTION: We are willing to replicate and extend these data in a new cohort of animals, by quantifying not only the lack of expression in astrocytes (using GFAP as well as S100b immunofluorescence) and in PV- and NPY-positive interneurons, but also by verifying it in other GABAergic populations (e.g., SOM) and all GABAergic interneurons (using GAD65-67 immunofluorescence). These data will help to shed more light on mixed results obtained so far with the use of this specific promoter.

5. Could the observed effect be a mere delay in epileptogenesis? A longitudinal study is crucial to explore this possibility, as the two-week recording period appears short given the dynamics shown in Figure 6E.

RE: This is indeed a possibility, even if our recording period was 3 and not 2 weeks. As described above (our response to Referee #1, point 6A), we did not prolong the recording period over 3 weeks because of limited battery duration and cost of the telemetry probe. Nonetheless, we are aware of the need of refining our observations in the future, in longer time periods and in other epilepsy models.

ACTION: We may add a comment in the Discussion stating the need to extend our finding in a longer timeframe of observation.

6. The use of T2A for NPY-Y2, as previously done by Melin et al., renders the discussion on this matter redundant. It would be more beneficial to focus on how this therapy fits into the context of other developed gene therapies.

RE: We respectfully disagree with the reviewer's comment. IRES, used by Melin et al. (2023) and T2A elements, used in the present work to simultaneously express two or more genes, are structurally different in terms of translational mechanism. We believe that is indeed relevant to discuss the topic not only with reference to our work, but also for potential future gene therapy products that might need to obtain the simultaneous expression of two genes.

7. Although the authors discussed the self-regulated approach, direct evidence supporting this claim is lacking in the present work.

RE: Since both transgenes are expressed by each single viral particle, they must be expressed together in each transduced cell. We have evidence of localization of both NPY and Y2R in mossy fiber terminals, i.e., of the anatomical substrate of autoregulation. In addition, we believe that the most parsimonious interpretation of the observation that NPY extracellular levels in vitro are increased much less by the vector expressing NPY and Y2R than by the vector expressing NPY alone is the existence of an autofeedback. This having said, we acknowledge that we need further verification that an auto-inhibitory pre-synaptic feedback loop on glutamate release can be established.

ACTION: We plan to perform experiments to show that neurons simultaneously over-expressing NPY and Y2 receptor release a lower amount of glutamate. Using cultured hippocampal neurons, transduced with either EGFP, NPY alone or Y2_NPY expressing vectors, we will i) directly measure the amount of glutamate released by neurons in which synaptic activity is stimulated (e.g., by high K⁺ or by treatment with 4-amino pyridine); ii) measure the vesicular release of glutamate, exploiting the properties of the FM1-43 dye (Amaral et al., Methods Mol Biol 2011), since the loading and loss of the fluorescent dye provide a reliable measurement of synaptic vesicle retrieval and release in cultured neurons (Sambri et al., EBioMedicine 2020).

8. If the authors aim to establish the superiority of this therapy over NPY alone, the addition of another group to the seizure analysis for a direct comparison is essential

RE: We are sorry that we did not make ourselves sufficiently clear on this point; we did not expect our approach to be superior to previous ones in terms of seizure control. Previous studies overexpressed NPY and/or Y2 in a non-cell specific, non-regulated manner, ensuring a constant and widespread increase of this inhibitory signal. We instead aimed at restricting expression to excitatory neurons (and mainly to a population of excitatory neurons, dentate gyrus granule cells), in an auto-regulated manner. Therefore, we wanted to test if this approach was sufficient to produce effects on seizures, not superior to other approaches.

ACTION: The manuscript will be revised to make this point clearer.

Referee #3 (Comments on Novelty/Model System for Author):

The manuscript describes the result of a newly developed lentiviral vector which enables co-expression of anti-convulsant neuropeptide NPY and its autoreceptor Y2 genes in the epileptic hippocampus, and evaluates its ability to suppress the seizures in vivo. The critical feature of this viral vector is its use of CamKIIa promotor driving both transgenes in the same excitatory neurons, which allows autoregulation of neural excitability through Y2 receptor-mediated inhibition. Telemetry video-EEG monitoring was conducted to detect seizures in the TKO mice expressing NPY/Y2 transgenes in dentate gyrus granule cells, and a clear reduction of the seizure occurrence in these mice was demonstrated.

This is a well thought out and seems to be a well carried out study. The idea of "autoregulation" of the epileptic brain activity is novel and very interesting. The choice of the model organism (i.e., mouse) is adequate, the statistical analysis methods used are adequate, and their usage of lentiviral vector for gene therapy seems to be reasonable and less hazardous compared to other viral vectors.

The main concern I have is that a few important control experiments are missing in this study. It weakens the overall impact of the core message "autoregulation of the epileptic activity". I

would like the authors discuss more thoroughly the mechanism how the autoregulation is realized in their model system to strengthen the core message.

RE: Since both transgenes are expressed by each single viral particle, they must be expressed together in each transduced cell. We present in vitro and in vivo evidence that this is indeed the case. In addition, we believe that the most parsimonious interpretation of the observation that NPY extracellular levels in vitro are increased much less by the vector expressing NPY and Y2R than by the vector expressing NPY alone is the existence of an autofeedback. This having said, we acknowledge that we need further verification that an auto-inhibitory pre-synaptic feedback loop on glutamate release can be established.

ACTION: Not only we will discuss more thoroughly the mechanism by which the autoregulation is realized, but we will also perform in vitro experiments to verify that neurons simultaneously over-expressing NPY and Y2 receptor release a lower amount of glutamate (see our response to Referee #2, point 7).

Major points;

1) "Necessity of combinatorial gene expression"

Authors demonstrated that expressing NPY and Y2R transgenes suppressed the seizures very well (Fig.6) and argue that the combinatorial expression is important. Unfortunately, however, the authors didn't show that expressing either of the two genes alone is NOT sufficient for suppressing seizures. Induction of the Y2R gene alone in the excitatory neurons may be sufficient for suppressing seizures, considering the low but constitutive expression of the endogenous NPY peptides in the hippocampus, and the increase in its expression level at the period of seizure onset (Fig. 5A). If that is the case, expressing a single gene, for example, a constitutively active form of the Y2R gene alone in place of the wild-type counterpart, seems to be a simpler and more secured way to suppress the seizures.

Forced expression of NPY gene in the excitatory neurons may also be sufficient. At least, authors could have examined the effect from NPY alone (LV_NPY) in vivo and evaluated its efficacy of suppression. I would like the authors to add a few control experiments to strengthen the argument that the combinatorial gene expression is critical.

RE: We apologize for not making ourselves sufficiently clear on the point that we did not expect our approach to be superior to previous ones in terms of seizure control. Previous studies overexpressed NPY and/or Y2 in a non-cell specific, non-regulated manner, ensuring a constant and widespread increase of this inhibitory signal. We instead aimed at restricting expression to excitatory neurons (and mainly to a population of excitatory neurons, dentate gyrus granule cells), in an auto-regulated manner. Therefore, we wanted to test if this approach was sufficient to produce effects on seizures, not superior to other approaches.

ACTION: We will rephrase key sentences in the manuscript to make our point clearer.

2) "Autoregulation on the seizure activity"

Authors argue that combinatorial expression of the NPY and Y2R genes in Hippocampal excitatory neurons exerts an "auto-feedback" regulation on the epileptic activity. When it comes to autoregulation, one would imagine a series of consecutive events in the hippocampus; 1) NPY release triggered by a spontaneous epileptic activity in the excitatory neurons, 2) Y2R activation by the induced NPY, 3) hyperpolarization of the membrane potential due to the K-channel opening after Y2R activation or whatever, and 4) premature

shut-down of the epileptic activity. If the scenario is right, frequency of the seizure events in TKO mice would not be influenced much by expressing the NPY-Y2R transgenes, but the seizure length should be shortened drastically. The results shown in figure 6 C,D, however, are not consistent with the scenario. Detectable seizure was nearly absent in TKO/LV_NPY_Y2R mice. I wonder how and when the NPY-Y2R system was activated without triggered by seizure activity in these seizure-free animals. I'd like the authors discuss more about the mechanism how the NPY-Y2R system autoregulates the spontaneous seizures in TKO mice. Another scenario is that TKO/LV_NPY_Y2R mice still had sub-threshold (undetectable) seizure activities frequently, which continuously activated the NPY/Y2R system in these mice. If that is the case, expressing a constitutively active form of the Y2R gene alone would be sufficient for suppressing the seizures.

RE: Existing data (see Krook-Magnuson et al., J Physiol 2015) support the notion (in keeping with the so called dentate 'gate' hypothesis) that preventing granule cell hyperexcitability can prevent occurrence of spontaneous seizures. Thus, our hypothesis is that an NPY-Y2R autoregulatory loop, by avoiding excessive granule cell activity and glutamate release, can prevent seizures. This implies that seizure frequency should be reduced whereas rare, occasional escape to control should evoke nearly "normal" seizures.

ACTION: We will discuss more carefully this hypothesis and the hypothesis outlined by the Referee. We may also add a graphical abstract like the one below.

3) "interference on the normal hippocampal function"

I agree with the authors that forced expression of the NPY-Y2R transgenes in the hippocampal excitatory neurons suppress the seizures quite well (Fig. 6). Now, I would like the authors to discuss the safety of the LV_NPY_Y2R.

It would be very nice if the authors can show that forced expression of the NPY-Y2R genes does not perturb the normal activity of the healthy excitatory neurons. Did authors observe any sign of behavioral abnormality in LV_NPY_Y2R mice? Authors should report detailed non-ictal behaviors/observations of the TKO/LV_NPY_Y2R mice. Autoregulating the epileptic activity as necessary seems to be definitely a better strategy than continuously lowering the neural activity using a constitutively active form of Y2R gene, which presumably has deleterious effects on the normal function of the hippocampus. Authors should discuss the merits and demerits of combinatorial gene expression strategy in comparison with single gene strategy, such as a constitutively active form of Y2R.

RE: We agree with the Referee that this is an important point.

ACTION: To test the safety of our LV platform, we will perform electrophysiological experiments on transduced primary hippocampal neurons and measure to which extent the overexpression of NPY-Y2, as compared with NPY alone, might influence normal neuronal activity. We will measure the basic electrophysiological properties of neurons and their spontaneous excitatory/inhibitory post-synaptic activity.

Minor points;

Fig.2 D. The bands for pre-processed NPY and mature NPY should be annotated and indicated by arrows. The unit (kDa ?) should also be indicated. The two lanes for LV_Y2-NPY are redundant.

RE: We thank the Referee for this suggestion.

ACTION: We will modify the image according to his/her suggestion.

Fig6 E,F. Did the authors also check whether the NPY/Y2R transgenes were successfully expressed or not in the TKO/LV_NPY_Y2R mice with seizures (n = 4)? Would the failure of suppressing the seizure in these mice be accounted for simply by a failure of NPY/Y2R transgene expression in the hippocampus?

RE: We thank the Referee for this valuable suggestion.

ACTION: We will extend the analysis on the brains of the animals employed in Figure 6 to verify the successful expression of transgenes.

Referee #3 (Remarks for Author):

The success of the combinatorial gene therapy using NPY-Y2R system is clear. The manuscript as in the current form is worth while publishing at least as a short report. For a full report, authors may want to consider the concerns above, and revise the manuscript.

RE: We thank the Referee for this comment, that we endorse completely. Indeed, we view this manuscript as the conceptual basis for future studies. We feel that the results are worthwhile communicating while embarking in further necessary but complex, long, and costly experiments.

Dear Michele,

Thank you for the transfer of your manuscript with referee reports to EMBO reports, and for your proposed revision plan.

I think your suggestions for how to revise your study are good, and I am happy to invite you to revise it with the understanding that the referee concerns must be fully addressed and their suggestions taken on board. Please address all referee concerns in a complete point-by-point response. Acceptance of the manuscript will depend on a positive outcome of a second round of review. It is EMBO reports policy to allow a single round of major revision only and acceptance or rejection of the manuscript will therefore depend on the completeness of your responses included in the next, final version of the manuscript.

We realize that it is difficult to revise to a specific deadline. In the interest of protecting the conceptual advance provided by the work, we recommend a revision within 3 months (27th Apr 2024). Please discuss the revision progress ahead of this time with the editor if you require more time to complete the revisions.

- 1) A data availability section providing access to data deposited in public databases is missing. If you have not deposited any data, please add a sentence to the data availability section that explains that.
- 2) Your manuscript contains statistics and error bars based on $n=2$. Please use scatter blots in these cases. No statistics should be calculated if $n=2$.

5) a complete author checklist, which you can download from our author guidelines . Please insert information in the checklist that is also reflected in the manuscript. The completed author checklist will also be part of the RPF.

6) Please note that all corresponding authors are required to supply an ORCID ID for their name upon submission of a revised manuscript (. Please find instructions on how to link your ORCID ID to your account in our manuscript tracking system in our Author guidelines

- the name of the statistical test used to generate error bars and P values,
- the number (n) of independent experiments (please specify technical or biological replicates) underlying each data point,
- the nature of the bars and error bars (s.d., s.e.m.),
- If the data are obtained from n {less than or equal to} 2, use scatter blots showing the individual data points.

I look forward to seeing a revised form of your manuscript when it is ready.

Cattaneo et al.

“Combinatorial gene therapy for epilepsy based on Neuropeptide Y and its Y2 receptor”
Ms number **EMBOR-2023-58691-T** (transfer from EMBO Mol Med EMM-2023-18970)

Point-by-point response to Reviewers

Referee #1 (Comments on Novelty/Model System for Author):

There are several papers showing an anticonvulsive effect of combined injections of NPY and Y2 vectors in other animal models. This paper, however uses CamKinase IIa promoters to allow expression only on glutamatergic neurons.

Referee #1 (Remarks for Author):

The authors use intrahippocampal injections of NPY and Y2 receptor vectors on CamKinase IIa promoter and test this in on epileptic activity in synapsin triple KO mice. This approach has been well established (several times) for NPY vectors and NPY + Y2 vectors in other animal models and using other promoters. The use of CamKinaseII promoters used here, however, has the purpose to restrict expression to glutamatergic neurons. Although the findings are not entirely novel, the paper shows a very nice analysis of the sites of vector expression and documents the anti-seizure effect of a combination of Y2 and NPY expressing vectors.

The experiments document expression of the vectors mainly (or exclusively) in mossy fibers. Surprisingly, they are obviously not expressed in mossy cells of the dentate gyrus. Why? Otherwise one would find NPY and Y2 in the inner molecular layer.

PROPOSAL OF RESPONSE TO EMBO REPORT EDITORS (approved with mail from Esther Schnapp dated January 26, 2024)

Indeed, some degree of expression in mossy cells was detected, with labeling of the inner molecular layer, which was more obviously detectable contralaterally. This information was reported and shown in the original manuscript (page 6, first 2 lines, and Fig. 3D).

ACTION: To refine this observation, we may perform immunofluorescence labeling of mossy cells in LV_EGFP animals using an antibody against mGluR2/3 (Bui et al., Science 2018).

RESPONSE

*We attempted to perform the mGluR2/3 staining using two commercially available antibodies and multiple experimental procedures, but failed to observe a specific signal with immunofluorescence, as per Bui et al. (2018). While we agree that this would have been a useful refinement, we trust that the original observation (**now at page 7, 2nd paragraph, lines 6-8, and Fig. 3D**) is sufficient for addressing the Reviewer's comment.*

Figure 5. What does the difference in NPY expression between B and D mean?

PROPOSAL OF RESPONSE

Panel B represents the quantitative measurements of NPY hippocampal levels that were performed in 5 animals per group per time point at four different timepoints (p30/60/90/120). Panel D is a representative image that qualitatively indicates the localization of NPY upregulation, that is, mainly in the mossy fiber terminals of the TKO mice.

RESPONSE

As above, in the proposal.

At p100 there was a high expression of NPY in mossy fibers (B), not at p120?

PROPOSAL OF RESPONSE

Indeed, we detected increased NPY expression at p90, when most of the animals were experiencing spontaneous seizures, while NPY levels returned to baseline by p120. In other words, increased NPY expression follows the onset of spontaneous seizures and precedes their attenuation. This suggests that NPY overexpression may represent an adaptive mechanism to seizure activity, that is set in place to counteract seizures themselves (which in fact decrease after p90 in TKO mice).

RESPONSE

As above, in the proposal.

How does NPY expression in mossy fibers relate to recent seizure activity?

PROPOSAL OF RESPONSE

In keeping with findings in other models (Vezzani et al., Neuropeptides 2004; Cattaneo et al., Front Mol Neurosci 2021), we hypothesize that reactive NPY overexpression reflects intensity of seizure activity also in the TKO model. However, we did not measure the correlation between seizure activity and NPY expression, as this was beyond the goals of the present study.

ACTION: This point may be briefly discussed in the revised manuscript.

RESPONSE

A sentence has been added as a comment to our finding of increased NPY levels at p90-p100 in the TKO model (page 9, 2nd paragraph, lines 2-4).

Is this expression typical for all animals?

PROPOSAL OF RESPONSE

Yes, this pattern is observed in all animals that were analyzed.

RESPONSE

As above, in the proposal.

Figure 6. To which time interval does "number of seizures" refer to? Video recording started two weeks after vector injection. How long did the recording last?

PROPOSAL OF RESPONSE

The time interval of "number of seizures" refers to the three weeks of recording.

ACTION: This may be better specified in the figure legend.

RESPONSE

A sentence has been added to specify this important point (page 9, 4th paragraph, lines 1-2).

6E. Would in the Y2-NPY group the number of seizure-free mice go further down after 30 days? I think one should investigate this.

PROPOSAL OF RESPONSE

We agree that it would be interesting to know whether seizure freedom is maintained for a longer time. Unfortunately, we could not prolong the recording period over 3 weeks because of limited duration of the telemetry battery. Please remember that the cost of recording animals in the telemetry system is very high. We are aware of the need of refining our observations in the future, in longer time periods and in other epilepsy models.

ACTION: We will explicitly state this need in the revised manuscript.

RESPONSE

A sentence has been added in the Discussion to mention this limitation of our study (page 14, 3rd paragraph, lines 3-5).

Referee #2 (Comments on Novelty/Model System for Author):

The choice of Synapsin TKO as model raises questions. It is not a recognized model of drug-resistant epilepsy, and its translational potential is unclear. The authors should have used other models where AAV-NPY-Y2 has already been tested

PROPOSAL OF RESPONSE

The multiple reasons why we selected the TKO model for this study are described in the original manuscript (page 6, last 3 lines to page 7, end of first paragraph). As stated above, while we are aware of the need of extending our observations to other models, we thought that this was a good model to start testing our new vector. Incidentally, please note that this point was raised only by Referee #2, whereas Referee #1 did not comment on it, and Referee #3 explicitly stated that "The choice of the model organism is adequate".

RESPONSE

As above, in the proposal. The statement on the reasons to select the TKO model can be found in the revised manuscript at (page 8, 2nd paragraph, lines 5 to end).

Referee #2 (Remarks for Author):

Cattaneo and colleagues devised a combinatorial approach involving lentivirus delivery of NPY and its receptor Y2, driven by a CamkII promoter. They demonstrated NPY and Y2 expression both in vitro and in vivo. Additionally, they observed no expression in astrocytes and inhibitory neurons in vivo. Subsequently, they administered this lentiviral tool to Synapsin TKO mice before the onset of seizures, revealing a reduction in seizures and an increase in seizure-free animals.

To the best of my knowledge, this marks the first gene therapy approach for Synapsin TKO mice. Nevertheless, caution is necessary before characterizing it as a "promising treatment option for drug-resistant epilepsy."

PROPOSAL OF RESPONSE

We agree with the Referee. Our statement in the Abstract was excessively optimistic, while our view is better expressed elsewhere in the manuscript.

ACTION: We will rephrase this sentence appropriately.

RESPONSE

The sentence has been rewritten (page 2, last 3 lines of the Abstract).

Several major concerns arise:

1. The statement, "However, none so far included a built-in auto-regulatory system, and each, therefore, may lead to excessive inhibition," is inaccurate. Previous auto-regulatory gene therapies for epilepsy, such as those conducted by Kullmann's lab at UCL, and those including work on NPY and Y2 co-expression using a single AAV (e.g., Melin et al., 2019, 2023, and Szczygie et al., 2020).

PROPOSAL OF RESPONSE

We agree that our statement is inaccurate. Please note, however, that whereas the Kullmann's lab (Qiu at al., Science 2022) proposed a regulated system based on the expression of therapeutic genes under the control of an activity-dependent promoter (that is, a regulation at transcriptional level), our approach entails an autoregulation in which transgenes are continuously expressed, not altering normal neuronal function but activating an auto-regulatory loop in case of hyper-activity. With reference instead to previous studies on NPY/Y2, as recognized by all Referees, the novelty is the expression of transgenes in excitatory neurons by using the CaMKII promoter (by the way, this is a necessary prerequisite for our auto-regulatory strategy). In sum, we believe that it is fair to say (words of Reviewer #3) that the study "can be another excellent work supporting the concept in a complementary fashion".

ACTION: We will revise the manuscript in order to make the novelty clearer, with reference in particular to the studies mentioned by the Referee.

RESPONSE

The statement mentioned by the Reviewers has been corrected (page 3, 3rd paragraph, lines 4-6). In addition, we rephrased several sentences in the 3rd paragraph of page 4 to emphasize the differences between our approach and those used previously.

2. The choice of Synapsin TKO as model raises questions. It is not a recognized model of drug-resistant epilepsy, and its translational potential is unclear. Why did the authors not opt for other models where AAV-NPY-Y2 has already been tested?

PROPOSAL OF RESPONSE

Please see above, our response to the Comments on Novelty of this Referee.

The multiple reasons why we selected the TKO model for this study are described in the original manuscript (page 6, last 3 lines to page 7, end of first paragraph). As stated

above, while we are aware of the need of extending our observations to other models, we thought that this was a good model to start testing our new vector. Incidentally, please note that this point was raised only by Referee #2, whereas Referee #1 did not comment on it, and Referee #3 explicitly stated that “The choice of the model organism is adequate”.

ACTION: We will explicitly state in the revised manuscript that it will be needed in the future to extend our observations to other models.

RESPONSE

As above, in the proposal. The statement on the reasons to select the TKO model can be found in the revised manuscript at page 8, 2ⁿ paragraph, lines 5 to end. A statement of the future need to extend the observations to other epilepsy models has also been added in the Discussion (page 14, 3rd paragraph, lines 3-6).

3. In terms of translation, lentiviral delivery in the human hippocampus may not be a realistic therapy due to the limited virus spread. Why was AAV not used, especially considering its previous use with NPY-Y2? The novelty here seems to lie in the specific excitatory promoter.

PROPOSAL OF RESPONSE

The choice of primarily using lenti- rather than AAV vectors was indeed based on the idea to use a specific excitatory promoter (CamKII), which implies a relatively large expression cassette that could hardly fit in AAV. In addition, we aimed at restricting the expression, for what possible, to granule cells. While AAV vectors certainly ensure a greater spread, one should also take into account that they must be used at high titers, which raise some concerns (e.g., BBB-breakdown and DNA interference - Guo et al., *Mol Ther Meth & Clin Devel* 2023). A comment of Referee #3 is perfectly in line with these considerations: “usage of lentiviral vector for gene therapy seems to be reasonable and less hazardous compared to other viral vectors”. Note also that a currently approved gene therapy clinical trial for focal epilepsy uses lentivectors (ClinicalTrials.gov ID NCT04601974); thus, it is not demonstrated that the limited spread of the virus cannot ensure a therapeutic effect.

ACTION: We may better explain the reasons for choosing a lentivector platform for the present study.

RESPONSE

We added a few sentences to explain the reasons why, in our experimental settings, a lentivector platform was preferable over AAV (page 4, 2nd paragraph, lines 3 to end).

4. The specificity of CamKII to excitatory neurons, particularly the lack of proper quantification, including inhibitory neurons and the percentage of transduced cells, needs clarification. Literature reports mixed results.

PROPOSAL OF RESPONSE

We have tested the specificity of CamKII in excitatory neurons by verifying lack of expression in either parvalbumin-positive and NPY-positive inhibitory neurons, and in astrocytes (Fig. 3).

ACTION: We are willing to replicate and extend these data in a new cohort of animals, by quantifying not only the lack of expression in astrocytes (using GFAP as well as

S100b immunofluorescence) and in PV- and NPY-positive interneurons, but also by verifying it in other GABAergic populations (e.g., SOM) and all GABAergic interneurons (using GAD65-67 immunofluorescence). These data will help to shed more light on mixed results obtained so far with the use of this specific promoter.

RESPONSE

Data have been extended and quantified (Results: page 7, 3rd paragraph; Methods: page 22, "Imaging and analysis"). A new figure has been added with a detailed analysis of EGFP expression in interneurons (Figure 4).

5. Could the observed effect be a mere delay in epileptogenesis? A longitudinal study is crucial to explore this possibility, as the two-week recording period appears short given the dynamics shown in Figure 6E.

PROPOSAL OF RESPONSE

This is indeed a possibility, even if our recording period was 3 and not 2 weeks. As described above (our response to Referee #1, point 6A), we did not prolong the recording period over 3 weeks because of limited battery duration and cost of the telemetry probe. Nonetheless, we are aware of the need of refining our observations in the future, in longer time periods and in other epilepsy models.

ACTION: We may add a comment in the Discussion stating the need to extend our finding in a longer timeframe of observation.

RESPONSE

A sentence has been added in the Discussion to mention this limitation of our study (page 14, 3rd paragraph, lines 3-6).

6. The use of T2A for NPY-Y2, as previously done by Melin et al., renders the discussion on this matter redundant. It would be more beneficial to focus on how this therapy fits into the context of other developed gene therapies.

PROPOSAL OF RESPONSE

We respectfully disagree with the reviewer's comment. IRES, used by Melin et al. (2023) and T2A elements, used in the present work to simultaneously express two or more genes, are structurally different in terms of translational mechanism. We believe that is indeed relevant to discuss the topic not only with reference to our work, but also for potential future gene therapy products that might need to obtain the simultaneous expression of two genes.

RESPONSE

As above, in the proposal. In addition, a more extensive description of how our gene therapy approach compares with other developed gene therapies has been added in the Introduction (page 4, last paragraph).

7. Although the authors discussed the self-regulated approach, direct evidence supporting this claim is lacking in the present work.

PROPOSAL OF RESPONSE

Since both transgenes are expressed by each single viral particle, they must be expressed together in each transduced cell. We have evidence of localization of both NPY and Y2R in mossy fiber terminals, i.e., of the anatomical substrate of autoregulation. In addition, we believe that the most parsimonious interpretation of the observation that NPY extracellular levels in vitro are increased much less by the vector expressing NPY and Y2R than by the vector expressing NPY alone is the existence of an autofeedback. This having said, we acknowledge that we need further verification that an auto-inhibitory pre-synaptic feedback loop on glutamate release can be established.

ACTION: We plan to perform experiments to show that neurons simultaneously over-expressing NPY and Y2 receptor release a lower amount of glutamate. Using cultured hippocampal neurons, transduced with either EGFP, NPY alone or Y2_NPY expressing vectors, we will i) directly measure the amount of glutamate released by neurons in which synaptic activity is stimulated (e.g., by high K⁺ or by treatment with 4-amino pyridine); ii) measure the vesicular release of glutamate, exploiting the properties of the FM1-43 dye (Amaral et al., *Methods Mol Biol* 2011), since the loading and loss of the fluorescent dye provide a reliable measurement of synaptic vesicle retrieval and release in cultured neurons (Sambri et al., *EBioMedicine* 2020).

RESPONSE

We performed the planned experiments on glutamate release, that are now described and discussed in the revised manuscript (Results: page 6, last paragraph ending at page 7; Methods, page 20; Figure 2G; Discussion: page 11, 1st paragraph, lines 9 to end). We also attempted to measure the vesicular release of glutamate using FM1-43. Unfortunately, this technique did not prove sufficiently specific and sensitive in our cultures. The reason for this unexpected outcome is unclear; the technique has been validated in peripheral neurons and embryonic cultures, but never tested in post-natal, central neuronal cultures like ours. While we believe that the level of evidence in favor of autoregulation is reasonable and maybe sufficient, we prudentially avoided to present it as solid and conclusive (Results: page 7, lines 3-4; Discussion: page 11, 1st paragraph, lines 9 to end).

8. If the authors aim to establish the superiority of this therapy over NPY alone, the addition of another group to the seizure analysis for a direct comparison is essential

PROPOSAL OF RESPONSE

We are sorry that we did not make ourselves sufficiently clear on this point; we did not expect our approach to be superior to previous ones in terms of seizure control. Previous studies overexpressed NPY and/or Y2 in a non-cell specific, non-regulated manner, ensuring a constant and widespread increase of this inhibitory signal. We instead aimed at restricting expression to excitatory neurons (and mainly to a population of excitatory neurons, dentate gyrus granule cells), in an auto-regulated manner. Therefore, we wanted to test if this approach was sufficient to produce effects on seizures, not superior to other approaches.

ACTION: The manuscript will be revised to make this point clearer.

RESPONSE

We revised the manuscript as described in the proposal. In particular, we added a few sentences in the Discussion to clarify this point (page 12, 1st paragraph).

Referee #3 (Comments on Novelty/Model System for Author):

The manuscript describes the result of a newly developed lentiviral vector which enables co-expression of anti-convulsant neuropeptide NPY and its autoreceptor Y2 genes in the epileptic hippocampus, and evaluates its ability to suppress the seizures in vivo. The critical feature of this viral vector is its use of CamKIIa promotor driving both transgenes in the same excitatory neurons, which allows autoregulation of neural excitability through Y2 receptor-mediated inhibition. Telemetry video-EEG monitoring was conducted to detect seizures in the TKO mice expressing NPY/Y2 transgenes in dentate gyrus granule cells, and a clear reduction of the seizure occurrence in these mice was demonstrated.

This is a well thought out and seems to be a well carried out study. The idea of "autoregulation" of the epileptic brain activity is novel and very interesting. The choice of the model organism (i.e., mouse) is adequate, the statistical analysis methods used are adequate, and their usage of lentiviral vector for gene therapy seems to be reasonable and less hazardous compared to other viral vectors.

The main concern I have is that a few important control experiments are missing in this study. It weakens the overall impact of the core message "autoregulation of the epileptic activity". I would like the authors discuss more thoroughly the mechanism how the autoregulation is realized in their model system to strengthen the core message.

PROPOSAL OF RESPONSE

Since both transgenes are expressed by each single viral particle, they must be expressed together in each transduced cell. We present in vitro and in vivo evidence that this is indeed the case. In addition, we believe that the most parsimonious interpretation of the observation that NPY extracellular levels in vitro are increased much less by the vector expressing NPY and Y2R than by the vector expressing NPY alone is the existence of an autofeedback. This having said, we acknowledge that we need further verification that an auto-inhibitory pre-synaptic feedback loop on glutamate release can be established.

ACTION: Not only we will discuss more thoroughly the mechanism by which the autoregulation is realized, but we will also perform in vitro experiments to verify that neurons simultaneously over-expressing NPY and Y2 receptor release a lower amount of glutamate (see our response to Referee #2, point 7).

RESPONSE

We performed the planned experiments on glutamate release, that are now described and discussed in the revised manuscript (Results: page 6, last paragraph ending at page 7; Methods, page 20; Figure 2G; Discussion: page 11, 1st paragraph, lines 9 to end).

Major points;

1) "Necessity of combinatorial gene expression"

Authors demonstrated that expressing NPY and Y2R transgenes suppressed the seizures very well (Fig.6) and argue that the combinatorial expression is important. Unfortunately, however, the authors didn't show that expressing either of the two genes alone is NOT sufficient for suppressing seizures. Induction of the Y2R gene alone in the excitatory neurons may be sufficient for suppressing seizures, considering the low but constitutive expression of

the endogenous NPY peptides in the hippocampus, and the increase in its expression level at the period of seizure onset (Fig. 5A). If that is the case, expressing a single gene, for example, a constitutively active form of the Y2R gene alone in place of the wild-type counterpart, seems to be a simpler and more secured way to suppress the seizures. Forced expression of NPY gene in the excitatory neurons may also be sufficient. At least, authors could have examined the effect from NPY alone (LV_NPY) in vivo and evaluated its efficacy of suppression. I would like the authors to add a few control experiments to strengthen the argument that the combinatorial gene expression is critical.

PROPOSAL OF RESPONSE

We apologize for not making ourselves sufficiently clear on the point that we did not expect our approach to be superior to previous ones in terms of seizure control. Previous studies overexpressed NPY and/or Y2 in a non-cell specific, non-regulated manner, ensuring a constant and widespread increase of this inhibitory signal. We instead aimed at restricting expression to excitatory neurons (and mainly to a population of excitatory neurons, dentate gyrus granule cells), in an auto-regulated manner. Therefore, we wanted to test if this approach was sufficient to produce effects on seizures, not superior to other approaches.

ACTION: We will rephrase key sentences in the manuscript to make our point clearer.

RESPONSE

We revised the manuscript as described in the proposal. In particular, we added a few sentences in the Discussion to clarify this point (page 12, 1st paragraph).

2) "Autoregulation on the seizure activity"

Authors argue that combinatorial expression of the NPY and Y2R genes in Hippocampal excitatory neurons exerts an "auto-feedback" regulation on the epileptic activity. When it comes to autoregulation, one would imagine a series of consecutive events in the hippocampus; 1) NPY release triggered by a spontaneous epileptic activity in the excitatory neurons, 2) Y2R activation by the induced NPY, 3) hyperpolarization of the membrane potential due to the K-channel opening after Y2R activation or whatever, and 4) premature shut-down of the epileptic activity. If the scenario is right, frequency of the seizure events in TKO mice would not be influenced much by expressing the NPY-Y2R transgenes, but the seizure length should be shortened drastically. The results shown in figure 6 C,D, however, are not consistent with the scenario. Detectable seizure was nearly absent in TKO/LV_NPY_Y2R mice. I wonder how and when the NPY-Y2R system was activated without triggered by seizure activity in these seizure-free animals. I'd like the authors discuss more about the mechanism how the NPY-Y2R system autoregulates the spontaneous seizures in TKO mice.

Another scenario is that TKO/LV_NPY_Y2R mice still had sub-threshold (undetectable) seizure activities frequently, which continuously activated the NPY/Y2R system in these mice. If that is the case, expressing a constitutively active form of the Y2R gene alone would be sufficient for suppressing the seizures.

PROPOSAL OF RESPONSE

Existing data (see Krook-Magnuson et al., J Physiol 2015) support the notion (in keeping with the so called dentate 'gate' hypothesis) that preventing granule cell hyperexcitability can prevent occurrence of spontaneous seizures. Thus, our

hypothesis is that an NPY-Y2R autoregulatory loop, by avoiding excessive granule cell activity and glutamate release, can prevent seizures. This implies that seizure frequency should be reduced whereas rare, occasional escape to control should evoke nearly "normal" seizures.

ACTION: We will discuss more carefully this hypothesis and the hypothesis outlined by the Referee. We may also add a graphical abstract like the one below.

RESPONSE

This hypothesis has been discussed (page 14, 2nd paragraph) and the graphical abstract has been added as Figure 8. Needless to say, should the Editor believe that this figure is redundant, we are ready to delete it or to add it to the manuscript as supplementary material.

3) "interference on the normal hippocampal function"

I agree with the authors that forced expression of the NPY-Y2R transgenes in the hippocampal excitatory neurons suppress the seizures quite well (Fig. 6). Now, I would like the authors to discuss the safety of the LV_NPY_Y2R.

It would be very nice if the authors can show that forced expression of the NPY-Y2R genes does not perturb the normal activity of the healthy excitatory neurons. Did authors observe any sign of behavioral abnormality in LV_NPY_Y2R mice? Authors should report detailed non-ictal behaviors/observations of the TKO/LV_NPY_Y2R mice. Autoregulating the epileptic activity as necessary seems to be definitely a better strategy than continuously lowering the neural activity using a constitutively active form of Y2R gene, which presumably has deleterious effects on the normal function of the hippocampus. Authors should discuss the merits and demerits of combinatorial gene expression strategy in comparison with single gene strategy, such as a constitutively active form of Y2R.

PROPOSAL OF RESPONSE

We agree with the Referee that this is an important point.

ACTION: To test the safety of our LV platform, we will perform electrophysiological experiments on transduced primary hippocampal neurons and measure to which extent the overexpression of NPY-Y2, as compared with NPY alone, might influence normal

neuronal activity. We will measure the basic electrophysiological properties of neurons and their spontaneous excitatory/inhibitory post-synaptic activity.

RESPONSE

We performed the planned experiments, which confirmed the absence of electrophysiological alterations in neurons transduced with LV_NPY or LV_Y2-NPY (Results: page 6, 3rd paragraph; Figure EV1; Methods: page 20).

Minor points;

Fig.2 D. The bands for pre-processed NPY and mature NPY should be annotated and indicated by arrows. The unit (kDa ?) should also be indicated. The two lanes for LV_Y2-NPY are redundant.

PROPOSAL OF RESPONSE

We thank the Referee for this suggestion.

ACTION: We will modify the image according to his/her suggestion.

RESPONSE

Changes in Figure 2D have been made.

Fig6 E,F. Did the authors also check whether the NPY/Y2R transgenes were successfully expressed or not in the TKO/LV_NPY_Y2R mice with seizures (n = 4)? Would the failure of suppressing the seizure in these mice be accounted for simply by a failure of NPY/Y2R transgene expression in the hippocampus?

PROPOSAL OF RESPONSE

We thank the Referee for this valuable suggestion.

ACTION: We will extend the analysis on the brains of the animals employed in Figure 6 to verify the successful expression of transgenes.

RESPONSE

As agreed, we extended analysis to all brains. Indeed, there may be a slightly lower expression of at least Y2R in animals that were having seizures. While this would reinforce the concept of efficacy of the treatment, we believe that the numbers are too low to reach firm conclusions. Therefore, we modified the paragraph only to specify that all brains were analyzed (page 10, 2nd paragraph), but would be reluctant to add this other piece of information.

Referee #3 (Remarks for Author):

The success of the combinatorial gene therapy using NPY-Y2R system is clear. The manuscript as in the current form is worth while publishing at least as a short report. For a full report, authors may want to consider the concerns above, and revise the manuscript.

PROPOSAL OF RESPONSE

We thank the Referee for this comment, that we endorse completely. Indeed, we view this manuscript as the conceptual basis for future studies. We feel that the results are

worthwhile communicating while embarking in further necessary but complex, long, and costly experiments.

RESPONSE

As above.

Dear Michele,

Thank you for the submission of your revised manuscript and also for your comments on the referee reports. I paste the reports again below. I understand your arguments and I agree with you that delaying the publication of your work by another 6 months is not a viable solution now. I would therefore like to invite you to address all final referee comments in the ms text and in a final point-by-point response. Please also rewrite the ms title and abstract to clearly point out the main advance your study provides. The description of your findings in the abstract needs to be written in present tense, as per journal policy.

A few editorial requests will also need to be addressed before we can proceed with the official acceptance of your manuscript:

- Please remove the author credits from the ms file. All credits need to be entered in our online ms submission system.
- Please add callouts for the individual panels of Figure 4 and Dataset EV1. These are currently missing.
- Dataset EV1 is a regular table, please re-name it Table 1 or Table EV1 and correct both the file name and the callouts in the ms text.
- The manuscript sections should be in the following order: Title page - Abstract & Keywords - Introduction - Results - Discussion - Methods - Data Availability - Acknowledgments - Disclosure Statement & Competing Interests - References - Figure Legends - (Main Tables with legends) - Expanded View Figure Legends.
- Materials and Methods should be called Methods
- Please note that the legends for figures 4b-g" are not provided in a sequential manner (legend for figures 4c-c", e-e", g-g" are provided before legend of figures 4b, d, f). This needs to be rectified.
- Please note that the exact p values are not provided in the legends of figures 2e-g; 5g; 6b; 7c-d.
- Please note that the error bars are not defined in the legend of figure 5g.

EMBO press papers are accompanied online by A) a short (1-2 sentences) summary of the findings and their significance, B) 2-3 bullet points highlighting key results and C) a synopsis image that is exactly 550 pixels wide and 200-600 pixels high (the height is variable). The synopsis image should provide a sketch of the major findings, like a graphical abstract. Please note that text needs to be readable at the final size. Please send us this information along with the final manuscript.

Referee #1:

I greatly appreciate the responses to my discussion points, which have significantly strengthened the manuscript. However, the model used, the evidence for autoregulatory mechanisms, and the translatability of the viral vector remain weaknesses. While lentivirus may be effective for small cortical malformations in humans, it is less likely to work in the hippocampus. I have two minor points to add:

1. Autoregulatory mechanisms: The paper I referenced is Lieb et al. (2018) in Nature Medicine, where an autoregulatory gene therapy based on glutamate levels was shown to be effective for focal epilepsy.
2. hCAMK2A specificity: I appreciate the new analysis on promoter specificity, and the results are intriguing. It might be beneficial to include a discussion of studies using AAVs, which have shown different results. This addition would help differentiate between lentivirus and AAV transduction.

Referee #2:

The authors use a mouse model of triple synapsin KO mice showing seizures. They inject lentiviral NPY/Y2 vectors on the CamKinasell promoter (targeting vector expression in principal neurons) for antagonizing seizures. They carefully characterize the targeting of the vectors in control mice (eg. expression on mossy fibers) and characterize the anti-convulsive effect.

Major comments:

A considerable drawback of the model is that the time period when seizures occur is obviously limited. Unfortunately, the authors performed their experiments (vector injections) not at the period of maximal seizures but the time before:

Thus, the maximal increase in endogenous NPY occurs P90 (Fig. 6 B') indicating that this was the time of maximal seizure activity. Telemetric recording (shown in Fig. 7) started on P60 (Fig. 7A) and lasted for 19 days (until P 79; Fig. 7E). The untreated mice had no seizures at the beginning; at the end of this period, practically all of them developed seizures. By this, around 60% of seizures were prevented by the LV-NPY-Y2 (lentivirus-NPY-Y2 receptor) injection (Fig. 7F). The authors performed their experiments when the animals show sub-maximal spontaneous seizures. In my opinion the interesting time period, however, would have been the time just afterwards (P75 to P100) when seizure activity was highest (Fig. 6B'). The authors only had to start their experiment (according to the results shown in Fig. 6B) two weeks later. It is interesting that the spontaneous seizures decline afterwards (P120; Fig 6 B'). Why is this so?

Minor Comments:

1. Fig. 7D: Mean seizure duration: this is somewhat misleading since only mice with seizures are included. This should be mentioned.
2. Fig. 8: The situation in the epileptic brain is not correctly shown. It implies that all NPY neurons degenerate. After status epilepticus somatostatin neurons degenerate in the tip of the hilus. Loss in NPY neurons is rather limited. In contrast, most NPY neurons up-regulate the neuropeptide (and GADs). This has been frequently shown (see also Fig. 6B of this paper). The paper referenced to (Housko et al) refers to a model of traumatic brain injury (TBI) in which 40 % of NPY neurons degenerated. This model is characterized by severe general brain damage (interneurons and principal neurons), although the rats show no epilepsy but only a reduced seizure threshold (presumably due to this general damage, including PV neurons). After electrical stimulation of the amygdala (inducing a status epilepticus) around 35 % of NPY degenerated in the hilus. Thus, it is not correct to refer to "loss of NPY + interneurons". Depending on the model the majority of interneurons survive. GABA-ergic terminal of the "healthy brain": NPY is co-stored with GABA in large dense core vesicles, GABA is contained also (without neuropeptides) in small synaptic vesicles.

I would include some explanations into legend (notably concerning the third neuron below).

2. Mention in the Methods section when you started the recording and how long it lasted
3. Legend to Fig 2: F-F': I would refer to NPY as peptide or neuropeptide not as protein.

In Conclusion:

1. The manuscript is not entirely novel: Combination of NPY and Y2 vectors have been shown before to have anticonvulsive effects (as NPY vectors alone).
2. The animal model may not be optimal. In my opinion it has also not been use optimal.
3. The use of a CamKinasell promoter for these vectors to selectively target principal neurons is novel
4. Characterization of vector targeting and vector expression is perfect
5. Also characterization of the animal model and of seizures is perfect

Referee #3:

Rev #3, Comments on Cattaneo et al., revision

The authors now provide a revision with additional necessary control experiments (Fig. 2G, Fig.EV1). I'm pleased to read the revision. However, I still have questions and comments on the revision. I'd like the authors to consider the comments below and reflect them in the revision.

1) Fig 2G.

The authors removed Mg²⁺ from the medium to increase the excitability of the hippocampal primary neuron culture. I don't quite understand the principle of the method. Please provide with relevant references. Independent experimental confirmation of a hyperexcitation by this manipulation may be necessary.

2) Fig EV1

The authors demonstrated that electrophysiological properties of isolated neurons transfected with the transgenes seem to be normal. However, consequences of introducing NPY-Y2R transgenes may become apparent only when the transfected neurons are integrated in the hippocampal neural circuits and operating there. Have authors monitored the wild type mouse injected with LV_NPY_Y2 vector (Figure 5) for their health and behavior? Any abnormality in, e.g., body weight increase, aggressiveness, feeding behavior, sleep-awake cycle, EEG patterns?

3) Practical merit of Lenti-virus vector.

In addition to the minimal spared of the virus vector in the brain tissue, I would also like the authors mention about the superior stability of the Lentivirus vector. These vectors are inserted into the chromosomes, once integrated, work permanently unlike other episomal AAV vectors which would be eventually lost in the cytoplasm. Lenti-virus vectors also have a high affinity to stem cells. Gene therapy with Lenti-vectors, therefore, eliminates the necessity of repeated injections into the body.

4) Adequacy of TKO mice as an epilepsy model animal (related to rev#2 comment).

As stated in the manuscript, it has been reported that the GABAergic signaling is impaired in TKO mice, and the failure of inhibition is the primary cause of the epileptic phenotype seen in these mutant mice. In this point, TKO mice is an adequate epilepsy model. However, I'd like to also remind the authors of the fact that synapsin II is expressed in excitatory neurons, therefore, glutamatergic synaptic signaling pathway is also impaired in these mice. In this regard, presentation of the new figure 8 needs a caution. Especially, in the middle panel titled "epileptic brain", the glutamatergic terminal is drawn as if it is operating normally. However, in the TKO mice used in this study, formation of the neurotransmitter vesicle pool at the terminal is presumably impaired. It is even possible that the NPY_Y2R auto-inhibition works well only in these synapsin II-missing excitatory synapses.

I'd like the authors to mention about this possibility in discussion.

5) Fig.2 D

A label for the molecular size unit ("kDa") is still missing (in other similar figures as well). The unit should be spelled out somewhere in the figure or mentioned in the legend.

6) Fig.2 G

This new experiment needs a control condition in which the basal glutamate release is measured in the normal medium (Mg2+ present), to demonstrate that the Mg2+ manipulation really induced a Glutamate release.

Cross-comments from referee 1:

I still believe that the model used (not translatable to human pathology, no patients with TKO for synapses) is a big weakness of this manuscript. And I agree with Ref#2 that also the time of intervention is not relevant to a potential gene therapy. So, I agree that showing an effect at the chronic phase of the pathology can have a more translational impact.

Cross-comments from referee 2:

The work mainly replicates that of Ledri et al and of others using a different vector and a different animal model. I think per se this can be published. The present work, however, suffers from the lack of demonstration of a long-time effect and from the use of a sub-optimal timing. In Ledri's work the effect on seizure development has been well demonstrated.

I agree with Referee 1. I think it is crucial to investigate the optimal time period (chronic phase). The additional experiment would also extend the whole time frame.

Cross-comments from referee 3:

It is unfortunate that the authors ended the recording (~P81), before the period of peak seizure activity (~P90). However, the anti-seizure effect is clearly demonstrated (Fig7E), even in the recording period. As the significant difference has been properly demonstrated between the age(day)-matched animals with/without Y2-NPY transgenes, I'm fine without a new experiment to be conducted at the period of maximal seizures.

Response to Editor and Referees comments

Editor

Thank you for the submission of your revised manuscript and also for your comments on the referee reports. I paste the reports again below. I understand your arguments and I agree with you that delaying the publication of your work by another 6 months is not a viable solution now. I would therefore like to invite you to address all final referee comments in the ms text and in a final point-by-point response. Please also rewrite the ms title and abstract to clearly point out the main advance your study provides. The description of your findings in the abstract needs to be written in present tense, as per journal policy.

The abstract has been rewritten as indicated.

A few editorial requests will also need to be addressed before we can proceed with the official acceptance of your manuscript:

Please remove the author credits from the ms file. All credits need to be entered in our online ms submission system.

The credits have been removed.

Please add callouts for the individual panels of Figure 4 and Dataset EV1. These are currently missing.

The callouts have been added at page 7 and 17

Dataset EV1 is a regular table, please re-name it Table 1 or Table EV1 and correct both the file name and the callouts in the ms text.

We re-named and inserted the Table as Table 1 at the bottom of the main manuscript file, corrected the callouts.

The manuscript sections should be in the following order: Title page - Abstract & Keywords - Introduction - Results - Discussion - Methods - Data Availability - Acknowledgments - Disclosure Statement & Competing Interests - References - Figure Legends - (Main Tables with legends) - Expanded View Figure Legends.

We reordered the manuscript sections.

Materials and Methods should be called Methods

We renamed the section.

Please note that the legends for figures 4b-g" are not provided in a sequential manner (legend for figures 4c-c", e-e", g-g" are provided before legend of figures 4b, d, f). This needs to be rectified.

We corrected the legend for figure 4.

Please note that the exact p values are not provided in the legends of figures 2e-g; 5g; 6b; 7c-d.

We provided the exact p values where requested.

Please note that the error bars are not defined in the legend of figure 5g.

The error bars are now defined in figure 5g

EMBO press papers are accompanied online by A) a short (1-2 sentences) summary of the findings and their significance, B) 2-3 bullet points highlighting key results and C) a synopsis image that is exactly 550 pixels wide and 200-600 pixels high (the height is variable). The synopsis image should provide a sketch of the major findings, like a graphical abstract. Please note that text needs to be readable at the final size. Please send us this information along with the final manuscript.

Here and attached we provide the short summary, key results and attached the synopsis image.

Summary of the findings and their significance

We designed a lentiviral vector co-expressing NPY with its inhibitory receptor Y2 in excitatory hippocampal cells, that allows inhibitory autoregulation of glutamate release. We demonstrate that efficient and selective overexpression of both genes in granule cell mossy fiber terminals is sufficient to induce a dramatic reduction in the frequency and duration of seizures in the synapsin triple KO model of epilepsy.

Key results

- Development of a LV vector with minimal CamKinasell promoter driving expression of NPY and its receptor Y2
- Efficient and nearly selective *in vivo* overexpression of NPY and Y2 receptor in granule cell mossy fiber terminals following vector administration in the dentate gyrus
- Reduction in the frequency and duration of seizures in the synapsin triple KO model of epilepsy

Referee 1

I greatly appreciate the responses to my discussion points, which have significantly strengthened the manuscript. However, the model used, the evidence for autoregulatory mechanisms, and the translatability of the viral vector remain weaknesses. While lentivirus may be effective for small cortical malformations in humans, it is less likely to work in the hippocampus. I have two minor points to add: 1. Autoregulatory mechanisms: The paper I referenced is Lieb et al. (2018) in Nature Medicine, where an autoregulatory gene therapy based on glutamate levels was shown to be effective for focal epilepsy.

We thank the reviewer for this annotation, we added a reference to Lieb et al. in the introduction at page 5.

2. hCAMK2A specificity: I appreciate the new analysis on promoter specificity, and the results are intriguing. It might be beneficial to include a discussion of studies using AAVs, which have shown different results. This addition would help differentiate between lentivirus and AAV transduction.

We thank the reviewer for this suggestion, we added some observations about this in the discussion at page 12 and 13.

Referee 2

The authors use a mouse model of triple synapsin KO mice showing seizures. They inject lentiviral NPY/Y2 vectors on the CamKinasell promoter (targeting vector expression in principal neurons) for antagonizing seizures. They carefully characterize the targeting of the vectors in control mice (eg. expression on mossy fibers) and characterize the anti-convulsive effect.

Major comments:

A considerable drawback of the model is that the time period when seizures occur is obviously limited. Unfortunately, the authors performed their experiments (vector injections) not at the period of maximal seizures but the time before:

Thus, the maximal increase in endogenous NPY occurs P90 (Fig. 6 B') indicating that this was the time of maximal seizure activity. Telemetric recording (shown in Fig. 7) started on P60 (Fig. 7A) and lasted for 19 days (until P 79; Fig. 7E). The untreated mice had no seizures at the beginning; at the end of this period, practically all of them developed seizures. By this, around 60% of seizures were prevented by the LV-NPY-Y2 (lentivirus-NPY-Y2 receptor) injection (Fig. 7F). The authors performed their experiments when the animals show sub-maximal spontaneous seizures. In my opinion the interesting time period, however, would have been the time just afterwards (P75 to P100) when seizure activity was highest (Fig. 6B'). The authors only had to start their experiment (according to the results shown in Fig. 6B) two weeks later.

It is interesting that the spontaneous seizures decline afterwards (P120; Fig 6 B'). Why is this so?

We thank the reviewer for this observation. However, we would like to point out that we actually applied our treatment at the time of peak seizure activity. We realize that we may not have made ourselves clear for the rationale of using the time point that we employed in the study, and this may have been misleading to the readers. Based on the literature, the susceptibility to seizures (that is, sensitivity to a seizure-provoking maneuver) in this model reaches a peak between P60 and P80 (Cambiaghi et al., Epil Res 2013 and at page 9). This is exactly the time window that we chose for 24/7 (24 h per day, 7 day a week) video-EEG monitoring. In fact, and in line with these findings, we observed spontaneous seizures in the control group within this time window. Seizures were observed in all the time window and did not display any tendency to increase in frequency, severity or duration in time (see the figure below). In particular, we would like to specify that the concept of "time to first seizure" (Fig. 7E) refers to the first occurrence of a seizure after a treatment, not in life. Regarding the peak of NPY expression (P90) that we report in Fig. 6, as we describe in the original manuscript (page 9, second paragraph), literature data suggest that increased NPY expression occurs as a reaction to seizure activity, indicating that the highest seizure activity in our model occurs before P90. This is the reason why we decided to anticipate its overexpression, ensuring that NPY is overexpressed while animals are in chronic epilepsy period.

We rewrote the paragraph at page 9 in a manner that should be clearer to the readers, by correcting a few wrong or misleading sentences.

The reason for the bell-shaped behavior in seizure frequency in TKO animals is not

clear, however the decline in spontaneous seizure activity in the TKO mice after P120 could be the result of the development of compensatory mechanisms, involving synaptic mechanisms differentially affecting excitatory and inhibitory neurons, during the chronicization of the epileptic phenotype.

Figure for referees not shown.

Minor Comments:

1. Fig. 7D: Mean seizure duration: this is somewhat misleading since only mice with seizures are included. This should be mentioned.

We thank the reviewer for the observation, we corrected the figure legend at page 28.

2. Fig. 8: The situation in the epileptic brain is not correctly shown. It implies that all NPY neurons degenerate. After status epilepticus somatostatin neurons degenerate in the tip of the hilus. Loss in NPY neurons is rather limited. In contrast, most NPY neurons up-regulate the neuropeptide (and GADs). This has been frequently shown (see also Fig. 6B of this paper). The paper referenced to (Housko et al) refers to a model of traumatic brain injury (TBI) in which 40 % of NPY neurons degenerated. This model is characterized by severe general brain damage (interneurons and principal neurons), although the rats show no epilepsy but only a reduced seizure threshold (presumably due to this general damage, including PV neurons). After electrical stimulation of the amygdala (inducing a status epilepticus) around 35 % of NPY degenerated in the hilus. Thus, it is not correct to refer to "loss of NPY + interneurons". Depending on the model the majority of interneurons survive. GABA-ergic terminal of the "healthy brain": NPY is co-stored with GABA in large dense core vesicles, GABA is contained also (without neuropeptides) in small synaptic vesicles. I would include some explanations into legend (notably concerning the third neuron below).

We thank the reviewer for the suggestion, we included a legend for figure 8 taking into account all these observations (page 29).

2. Mention in the Methods section when you started the recording and how long it lasted

We modified the methods section, at page 24.

3. Legend to Fig 2: F-F': I would refer to NPY as peptide or neuropeptide not as protein.

We modified the legend, at page 26.

In Conclusion:

1. The manuscript is not entirely novel: Combination of NPY and Y2 vectors have been shown before to have anticonvulsive effects (as NPY vectors alone).
2. The animal model may not be optimal. In my opinion it has also not been use optimal.
3. The use of a CamKinasell promoter for these vectors to selectively target principal neurons is novel
4. Characterization of vector targeting and vector expression is perfect
5. Also characterization of the animal model and of seizures is perfect

Referee 3

Rev #3, Comments on Cattaneo et al., revision

The authors now provide a revision with additional necessary control experiments (Fig. 2G, Fig.EV1). I'm pleased to read the revision. However, I still have questions and comments on the revision. I'd like the authors to consider the comments below and reflect them in the revision.

1) Fig 2G.

The authors removed Mg²⁺ from the medium to increase the excitability of the hippocampal primary neuron culture. I don't quite understand the principle of the method. Please provide with relevant references. Independent experimental confirmation of a hyperexcitation by this manipulation may be necessary.

The incubation of cultured neurons or tissue slices in a Mg²⁺-free solution, allowing the activation of NMDA receptors due to the removal of the Mg²⁺ blockade of the channel, is a very common stimulus used to induce in vitro a SE-like condition (see for example Hogg et al. 2019, Mele et al. 2021, Mangan et al. 2004). The references were added in the results section, page 7. The figure below (now added to the manuscript as Figure EV2) shows that Mg²⁺ removal increases glutamate release in neurons transduced with LV_EGFP.

2) Fig EV1

The authors demonstrated that electrophysiological properties of isolated neurons transfected with the transgenes seem to be normal. However, consequences of introducing NPY-Y2R transgenes may become apparent only when the transfected neurons are integrated in the hippocampal neural circuits and operating there. Have authors monitored the wild type mouse injected with LV_NPY_Y2 vector (Figure 5) for their health and behavior? Any abnormality in, e.g., body weight increase, aggressiveness, feeding behavior, sleep-awake cycle, EEG patterns?

The observation of the reviewer is obviously correct. We did not monitor the animals' health by using specific tests. However, we did not observe any change in their gross behavior (including aggressiveness and feeding behavior), weight curve and EEG patterns (including that during sleep). We added this info in the Results section (page 8).

3) Practical merit of Lenti-virus vector.

In addition to the minimal spared of the virus vector in the brain tissue, I would also like the authors mention about the superior stability of the Lentivirus vector. These vectors are inserted into the chromosomes, once integrated, work permanently unlike other episomal AAV vectors which would be eventually lost in the cytoplasm. Lenti-virus vectors also have a high affinity to stem cells. Gene therapy with Lenti-vectors, therefore, eliminates the necessity of repeated injections into the body.

We thank the reviewer for the observation, we added a sentence at page 4.

4) Adequacy of TKO mice as an epilepsy model animal (related to rev#2 comment).

As stated in the manuscript, it has been reported that the GABAergic signaling is impaired in TKO mice, and the failure of inhibition is the primary cause of the epileptic phenotype seen in these mutant mice. In this point, TKO mice is an adequate epilepsy model. However, I'd like to also remind the authors of the fact that synapsin II is expressed in excitatory neurons, therefore, glutamatergic synaptic signaling pathway is also impaired in these mice. In this regard, presentation of the new figure 8 needs a caution. Especially, in the middle panel titled "epileptic brain", the glutamatergic terminal is drawn as if it is operating normally. However, in the TKO mice used in this study, formation of the neurotransmitter vesicle pool at the terminal is presumably impaired. It is even possible that the NPY_Y2R auto-inhibition works well only in these synapsin II-missing excitatory synapses. I'd like the authors to mention about this possibility in discussion.

We added this information at page 8 of the revised manuscript.

5) Fig.2 D

A label for the molecular size unit ("kDa") is still missing (in other similar figures as well). The unit should be spelled out somewhere in the figure or mentioned in the legend.

We modified the figures according to the reviewer's indication and added the molecular weight labeling.

6) Fig.2 G

This new experiment needs a control condition in which the basal glutamate release is measured in the normal medium (Mg²⁺ present), to demonstrate that the Mg²⁺ manipulation really induced a Glutamate release.

As said, the incubation of cultured neurons or tissue slices in a Mg²⁺-free solution, allowing the activation of NMDA receptors due to the removal of the Mg²⁺ blockade of the channel, is a very common stimulus used to induce in vitro a SE-like condition (see for example Hogg et al. 2019, Mele et al. 2021, Mangan et al. 2004). The figure below (now added to the manuscript as Figure EV2) shows that Mg²⁺ removal increases glutamate release in neurons transduced with LV_EGFP.

Cross-comments from referee 1:

I still believe that the model used (not translatable to human pathology, no patients with TKO for synapses) is a big weakness of this manuscript. And I agree with Ref#2 that also the time of intervention is not relevant to a potential gene therapy. So, I agree that showing an effect at the chronic phase of the pathology can have a more translational impact.

Cross-comments from referee 2:

The work mainly replicates that of Ledri et al and of others using a different vector and a different animal model. I think per se this can be published. The present work, however, suffers from the lack of demonstration of a long-time effect and from the use of a sub-optimal timing. In Ledri's work the effect on seizure development has been well demonstrated.

I agree with Referee 1. I think it is crucial to investigate the optimal time period (chronic phase). The additional experiment would also extend the whole time frame.

Cross-comments from referee 3:

It is unfortunate that the authors ended the recording (~P81), before the period of peak seizure activity(~P90). However, the anti-seizure effect is clearly demonstrated (Fig7E), even in the recording period. As the significant difference has been properly

demonstrated between the age(day)-matched animals with/without Y2-NPY transgenes, I'm fine without a new experiment to be conducted at the period of maximal seizures.

RESPONSE to the Cross Comments

We would like to point out that we actually applied our treatment at the time of peak seizure activity. We realize that we may not have made ourselves clear for the rationale of using the time point that we employed in the study, and this may have been misleading to the readers. Thus, the paragraph at page 9 has been extensively rewritten. Based on the literature, the susceptibility to seizures (that is, sensitivity to a seizure-provoking maneuver) in this model reaches a peak between P60 and P80 (Cambiaghi et al., Epil Res 2013 and at page 9). This is exactly the time window that we chose for 24/7 (24 h per day, 7 day a week) video-EEG monitoring. In fact, and in line with these findings, we observed spontaneous seizures in the control group within this time window. Seizures were observed in all the time window and did not display any tendency to increase in frequency, severity or duration in time (see the figure below). In particular, we would like to specify that the concept of "time to first seizure" (Fig. 7E) refers to the first occurrence of a seizure after a treatment, not in life. Regarding the peak of NPY expression (P90) that we report in Fig. 6, as we describe in the manuscript (page 9, second paragraph), literature data suggest that increased NPY expression occurs as a reaction to seizure activity, indicating that the highest seizure activity in our model occurs before P90. This is the reason why we decided to anticipate its overexpression, ensuring that NPY is overexpressed while animals are in chronic epilepsy period.

Regarding the translatability issue: based on the above, we believe that our experimental design is correct for testing effects in a chronic phase of epilepsy. Note also that our present work includes one of the longer-lasting video-EEG monitoring among those on NPY-Y2 gene therapy. There is no single animal model that is fully validated for translation in human epilepsy, except for antiseizure effects. While we acknowledge the limitations of the TKO model, we also recognize that it holds many interesting features that led us to choose it as a first testing model. Note that aim of this work was not to provide immediately translatable evidence for a novel approach. Admittedly, we stated something like that in a couple of sentences of the original manuscript, which was correctly noted by the Referees, leading us to a more accurate definition of the aims and a more conservative conclusive statement ("these results suggest that an auto-regulated gene therapy approach based on co-expression of NPY and Y2R in hippocampal GCs is feasible and worth further investigation"). In other words, the aim of our work was to provide initial evidence that the well-known NPY-Y2 approach to the gene therapy of epilepsy may be refined by targeting as specifically as possible a relatively small subset of neurons, instead of the widespread and non-specific approaches used so far.

Prof. Michele Simonato
IRCCS Ospedale San Raffaele
Milan
Italy

Dear Prof. Simonato,

I am very pleased to accept your manuscript for publication in the next available issue of EMBO reports. Thank you for your contribution to our journal.
